# Synthesizing Minority Samples for Long-tailed Classification via Distribution Matching

**Zhuo Li**                                                                                          *zhuoli3@link.cuhk.edu.cn*
*Shenzhen International Center for Industrial and Applied Mathematics,*
*Shenzhen Research Institute of Big Data,*
*The Chinese University of Hong Kong, Shenzhen,*

**He Zhao**                                                                                          *he.zhao@data61.csiro.au*
*CSIRO's Data61, Australia*

**Jinke Ren**                                                                                          *jinkeren@cuhk.edu.cn*
*Shenzhen Future Network of Intelligence Institute*
*School of Science and Engineering, The Chinese University of Hong Kong, Shenzhen*
*Guangdong Provincial Key Laboratory of Future Networks of Intelligence*

**Anningzhe Gao**                                                                                          *anningzhegao@gmail.com*
*Shenzhen Research Institute of Big Data*

**Dandan Guo**[*]                                                                                          *guodandan@jlu.edu.cn*
*Jilin University*

**Xiang Wan**                                                                                          *wanxiang@sribd.cn*
*Shenzhen Research Institute of Big Data*

**Hongyuan Zha**                                                                                          *zhahy@cuhk.edu.cn*
*The Chinese University of Hong Kong, Shenzhen*

**Reviewed on OpenReview:** *https://openreview.net/forum?id=VqLe8tPbZn*

## Abstract

In many real-world applications, deep neural networks (DNNs) often perform poorly on datasets with long-tailed distributions. To address this issue, a promising approach is to propose an optimization objective to transform real majority samples into synthetic minority samples. However, this objective is designed only from the classification perspective. To this end, we propose a novel framework that synthesizes minority samples from the majority by considering both classification and distribution matching. Specifically, our method adjusts the distribution of synthetic minority samples to closely align with that of the true minority class, while enforcing the synthetic samples to learn more generalizable and discriminative features of the minority class. Experimental results on several standard benchmark datasets demonstrate the effectiveness of our method in both long-tailed classification and synthesizing high-quality synthetic minority samples.

## 1 Introduction

The success of deep learning for supervised learning relies on high-quality large-scale datasets, which are often assumed to have nearly balanced numbers of samples for each class (Russakovsky et al., 2015). However, real-world datasets usually suffer from a long-tailed problem, where a few majority classes occupy most

---

[*]Corresponding Author. Code is available on `https://github.com/BIRlz/TMLR_Syn-LT`.

data while many minority classes have very few samples (Zhou et al., 2017; Liu et al., 2015). Deep neural networks (DNNs) trained on long-tailed datasets have poor generalization performance, especially in minority classes (Zhou et al., 2020; Liu et al., 2019). Therefore, it is of practical importance to develop methods for mitigating the long-tailed problem.

To alleviate the imbalance issue, several kinds of methods have been proposed in the past decade, in which the data-level approach has received significant attention due to the simplicity and effectiveness (Yang et al., 2022). This approach usually aims to achieve a balanced training data distribution via re-sampling (*i.e.*, under-sampling (He & Garcia, 2009), over-sampling (Van Hulse et al., 2007; Gao et al., 2023)) or data augmentation (Chu et al., 2020; Hong et al., 2022; Li et al., 2021; Ahn et al., 2023; Gao et al., 2024b). In the context of re-sampling, one representative method is the Synthetic Minority Over-sampling Technique (SMOTE) (Chawla et al., 2002), which synthesizes minority samples by interpolating between existing minority samples and their nearest neighbor samples. Recently, Kim et al. (2020) have revisited the over-sampling framework and proposed a new way of to synthesize minority samples, called Major-to-minor (M2m). A unique advantage of M2m over SMOTE-based methods is that M2m utilizes the majority samples to generate minority samples via an optimization process, thus can "cook with much more raw materials". In this way, M2m is able to "transform" majority samples into minority samples to achieve a balanced dataset.

Despite its initial success, M2m only focuses on optimizing the synthetic minority samples from the classification perspective, whose similarity with the real samples in the concerned minority class is overlooked. In other words, a synthetic sample $\hat{x}$ initialized from a majority class $k_0$ is viewed as a sample of the minority class $k$ if a pre-trained classifier $g$ identifies it as class $k$ confidently, and a target classifier $f$ has low confidence about it on $k_0$, ignoring whether the synthetic sample upholds the genuine characteristics (i.e., feature distribution) of the class $k$.

In this work, we propose a novel framework for synthesizing minority samples via distribution matching. Our insight is that *a desired synthetic minority sample should not only satisfy the classification constraints about $g(\hat{x})$ and $f(\hat{x})$, but also be distributionally close to real samples in the target minority class.* To satisfy these, we introduce a principled approach that optimizes the synthetic minority samples by enforcing them to satisfy the classification constraints and be close to the distribution of real samples, by minimizing the optimal transport (OT) distance (Peyré et al., 2019). Moreover, to mitigate the harmfulness of unreliable synthetic samples, we define a sample rejection criterion based on the distance between synthetic minority samples and real minority samples. Concurrently, the classifier $f$ is jointly trained leveraging these progressively refined synthetic samples, creating a synergistic learning process.

In order to enhance the generality and practical applicability of our method, we introduce an additional regularization term concerning the "confusing class" within the minority class, which accounts for instances where minority samples are frequently misclassified into a specific class rather than other classes. In this way, we relax the label requirement of majority samples and as a result, our proposed method can translate not only In-Distribution (ID) majority samples but also Out-of-Distribution (OOD) samples (Wei et al., 2022) into synthetic minority samples, making ours more applicable in practice. Additionally, our method can also be used as a plug-in approach to enhance the performance of other methods, e.g., reweighting loss. Moreover, we conduct extensive experiments on standard benchmark datasets and our method achieves improved long-tailed classification performance. In conclusion, our contributions are summarized as follows:

- To address the long-tailed classification problem, we propose a general framework for synthesizing minority samples via distribution matching, where we formulate real samples and synthetic ones as two distributions.

- We optimize synthetic minority samples by enforcing them to satisfy the classification constraints and keep close to the real representation distribution by minimizing the OT distance.

- We significantly enhance the framework's robustness and versatility by introducing two novel components: a an innovative regularization term focused on "confusing classes", which crucially enables effective synthesis from (ID) and (OOD) seed samples; and (b) an effective feature-distance-based sample rejection criterion to ensure the quality of generated instances.

- Extensive experiments on standard benchmarks demonstrate the effectiveness of our method, showing that is a promising over-sampling framework for the long-tailed classification problem.

## 2 Preliminaries

**Optimal transport.** OT is a widely used measurement for comparing distributions (Peyré et al., 2019), where we only focus on the discrete situation that is more related to our framework. Assuming we have two sets of points (features), we can formulate the discrete distributions as $P = \sum_{n=1}^{N} u_n \delta_{x_n}$ and $Q = \sum_{m=1}^{M} v_m \delta_{y_m}$, where $\delta$ is Dirac function, intuitively a unit of mass which is infinitely concentrated at location $x_i$. Dirac function helps convert a collection of raw data into a discrete probability distribution, by modeling each point as an individual "mass". $\boldsymbol{u} \in \Delta^N$ and $\boldsymbol{v} \in \Delta^M$ are the discrete probability vectors that sum to 1. The discrete OT distance between distribution $P$ and $Q$ can be formulated as:

$$\min_{\mathbf{T} \in \Pi(P,Q)} \langle \mathbf{T}, \mathbf{C} \rangle = \sum_{n}^{N} \sum_{m}^{M} T_{nm} C_{nm}, \tag{1}$$

where $\mathbf{C} \in \mathbb{R}_{>0}^{n \times m}$ is the cost matrix whose each point denotes the distance between $x_n$ and $y_m$ and transport probability matrix $\mathbf{T} \in \mathbb{R}_{>0}^{n \times m}$ satisfies $\Pi(P,Q) := \left\{ \mathbf{T} | \sum_{n=1}^{N} T_{nm} = v_m, \sum_{m=1}^{M} T_{nm} = u_n \right\}$. As directly optimizing 1 is always time-expensive, Sinkhorn algorithm (Cuturi, 2013) introduces an entropic constraint, *i.e.*, $H(\mathbf{T}) = -\sum_{nm} T_{nm} \ln T_{nm}$ for fast optimization.

**Long-tailed classification.** Assume a training dataset $\mathcal{D}_{\text{train}} = \{(x_i, y_i)\}_{i=1}^{N}$, where $x_i \in \mathbb{R}^d$ denotes the $i$-th input and $y_i$ means its corresponding label over $K$ classes. Let $N$ denote the number of the entire training data and $N_k$ is that of class $k$, where we assume $N_1 \geq N_2 \geq ... \geq N_K$ without loss of generality. Following Cao et al. (2019), we define the imbalance factor (IF) as $\frac{N_1}{N_K}$ which directly quantifies the ratio between the largest and smallest class sizes, providing a clear measure of dataset skewness. For example, when IF is larger, the training set is more imbalanced, i.e., more challenging. Denote $f : \mathbb{R}^d \to \mathbb{R}^K$ as the target classifier, which can be learned by empirical risk minimization (ERM) over the imbalanced training set with an appropriate loss function $\mathcal{L}(f)$:

$$\min_{f} \mathbb{E}_{(x,y) \sim \mathcal{D}_{\text{train}}}[\mathcal{L}(f; x, y)]. \tag{2}$$

However, when $\mathcal{D}_{\text{train}}$ exhibits a long-tailed distribution, the standard ERM objective in Eq. 2 inherently struggles to ensure fairness across classes. Because the expectation $\mathbb{E}_{(x,y) \sim \mathcal{D}_{\text{train}}}$ is dominated by the high-frequency majority classes, the learned model $f$ becomes heavily biased towards these head categories. This direct consequence of the ERM formulation on imbalanced data usually results in poor generalization performance on underrepresented tail classes (Fang et al., 2021; Han et al., 2023).

## 3 Method

### 3.1 Motivation

To motivate our method, we first review the most related work in the line of over-sampling, called Major-to-minor (M2m) (Kim et al., 2020). M2m aims to construct a new balanced dataset $\mathcal{D}_{\text{bal}}$ from the original dataset $\mathcal{D}_{\text{train}}$ by generating $N_1 - N_k$ synthetic samples for each class $k$, where the concerned classifier $f$ trained on $\mathcal{D}_{\text{bal}}$ is expected to perform better than that trained on $\mathcal{D}_{\text{train}}$. Note that $N_1 \geq N_2 \geq ... \geq N_K$. Therefore, generating $N_1 - N_k$ synthetic samples for each class $k$ enables a balanced dataset. Here, synthetic samples in minority classes are generated by translating from other samples in majority classes. In addition to the to-be-learned classifier $f$ trained on $\mathcal{D}_{\text{bal}}$, M2m assumes a baseline classifier $g$ pre-trained on the imbalanced dataset $\mathcal{D}_{\text{train}}$ with standard ERM training, where $f$ and $g$ have the same structure. Although $g$ may not achieve the optimal performance, it is expected to achieve reasonable performance on the imbalanced training dataset. M2m designs to obtain a synthetic sample $\hat{x}$ for a minority class $k$, which is translated from a real training sample $x_0$ from a major class $k_0$ in $\mathcal{D}_{\text{train}}$, through several optimized steps.

Although M2m can achieve promising results, it translates $\hat{x}$ from $x_0$ purely in the view of classification and ignores the similarity between $\hat{x}$ and the corresponding real samples in the concerned minority class $k$. In this scenario, the synthetic sample might mislead the model and cause inaccurate predictions.

### 3.2 Learning Synthetic Minority Samples with Distribution Matching

To address the above issue, we aim to learn high-quality synthetic samples that not only satisfy the classification constraints about $g(\hat{x})$ and $f(\hat{x})$ but also follow the distribution of real samples in the target minority class $k$. We achieve the first goal by solving the optimization problem within $T$ steps, as shown below:

$$\hat{x}^{(t)} = \arg\min_{\hat{x}} \mathcal{L}(g(\hat{x}), k) + \lambda f_{k_0}(\hat{x}), \tag{3}$$

where $\hat{x}$ is initialized with $x_0 + \epsilon$ and $\epsilon$ is standard Gaussian noise, i.e., $\hat{x}^{(0)} \leftarrow x_0 + \epsilon$. $\hat{x}^{(t)}$ denotes the synthetic sample at $t$-th optimization step within $T$ iterations. Next, we focus on the second goal. Taking the $k$-th class in $\mathcal{D}_{\text{train}}$ as an example, we denote $\mathcal{D}_k = \{(x_n, y_n)\}_{n=1}^{N_k}$ as the set of real samples, and $\hat{\mathcal{D}}_k = \{(\hat{x}_m, \hat{y}_m)\}_{m=1}^{M_k}$ as the to-be-learned synthetic set, where $M_k = N_1 - N_k$ is the number of synthetic samples of class $k$. Then the empirical distributions of $\mathcal{D}_k$ and $\hat{\mathcal{D}}_k$ can be formulated as:

$$P_k = \sum_{n=1}^{N_k} \frac{1}{N_k} \delta_{x_n}, \qquad Q_k = \sum_{m=1}^{M_k} \frac{1}{M_k} \delta_{\hat{x}_m}. \tag{4}$$

Note that label is omitted since $\hat{y}_m = y_n = k$. Moving beyond Eq. 3, which only utilizes the classification loss to learn minority synthetic samples, we further introduce a distribution matching loss to enforce the to-be-learned distribution $Q_k$ to stay close to the real distribution $P_k$ of class $k$. Let $\text{Dist}(P_k, Q_k)$ denote the distance between the distributions $P_k$ and $Q_k$. Here we adopt the principled approach of OT to define $\text{Dist}(P_k, Q_k)$, although other approaches are also available, such as Maximum Mean Discrepancy (MMD) (Gretton et al., 2012) and Energy Distance (ED) (Rizzo & Székely, 2016). We defer the implementation of $\text{Dist}(P_k, Q_k)$ with other measures such as Maximum Mean Discrepancy (MMD) (Gretton et al., 2012) and Energy Distance (ED) (Rizzo & Székely, 2016) to Appendix A.

Since the training images are high dimensional, minimizing the distribution distance in the image space is expensive and inaccurate. Therefore, we assume an embedding function $\psi_\theta : \mathbb{R}^d \to \mathbb{R}^{d'}$ parameterized with $\theta$ and compute the distribution distance $\text{Dist}_\theta(P_k, Q_k)$ in the feature space. Specifically, we define it using the entropic OT:

$$\text{Dist}_\theta(P_k, Q_k) = \min_{\mathbf{T} \in \Pi(P_k, Q_k)} \langle \mathbf{T}, \mathbf{C} \rangle - \gamma H(\mathbf{T}), \tag{5}$$

where $\gamma > 0$ is a hyper-parameter for the entropy constraint $H(\mathbf{T})$. The transport plan satisfies:

$$\Pi(P_k, Q_k) := \left\{ \mathbf{T} \mid \sum_{n=1}^{N_k} T_{nm} = 1/M_k, \sum_{m=1}^{M_k} T_{nm} = 1/N_k \right\}, \tag{6}$$

and the cost function $C_{nm}$ measures the distance between the real sample $x_n$ and synthetic sample $\hat{x}_m$. $C_{nm}$ can be viewed as a distance metric in the embedding space. Although theoretically it is possible to use any reasonable distance metric, we use the cosine similarity, *i.e.*, $C_{nm} = 1 - \cos(\psi_\theta(x_n), \psi_\theta(\hat{x}_m))$, which gives the best performance in this work.

### 3.3 Embedding Function $\psi_\theta$ and Optimization Problem

In order to achieve efficient computation of distribution distance, the parameterization of an embedding function $\psi_\theta$ is necessary and important. Commonly, we can employ the feature extractor in $g$ or $f$ as the embedding function. However, $g$ is a biased model that carries class-specific biases relevant to the long-tailed distribution and $f$ is learned during each training iteration, whose parameters may not be optimal for computing the $\text{Dist}_\theta(P_k, Q_k)$. In order to create diverse and unbiased projections in a computationally efficient manner, motivated by Zhao & Bilen (2023) that computes feature distance based on a family

of models, we also match $P_k$ and $Q_k$ in many sampled embedding spaces. Specifically, when computing $\text{Dist}_\theta(P_k, Q_k)$ each time, we employ a neural network that keeps the same architecture with the encoder of our target classifier $f$ to serve as our embedding function $\psi_\theta$, which thus can be initialized by commonly used Kaiming Initialization (He et al., 2015). Moreover, we experimentally validate that the family of randomly initialized embedding spaces can produce better results than using one embedding space.

To summarize, we can minimize $\mathbb{E}_{\theta \sim P_\theta}[\text{Dist}_\theta(P_k, Q_k)]$ such that the synthetic samples are optimized to match the original data distribution in various embedding spaces. The overall optimization objective is formulated as:

$$\hat{x}_m^{(t)} = \underset{\{\hat{x}_m\}_{m=1}^{M_k}}{\arg\min} \sum_{m=1}^{M_k} \left[ \mathcal{L}(g(\hat{x}_m), k) + \lambda_1 f_{k_0}(\hat{x}_m) \right] + \lambda_2 \mathbb{E}_{\theta \sim P_\theta}[\text{Dist}_\theta(P_k, Q_k)], \tag{7}$$

where each $\hat{x}_m$ is initialized by a randomly sampled image $x_m$ from a major class and $\epsilon_m$ is the randomly sampled Gaussian noise, i.e., $\hat{x}_m^{(0)} \leftarrow x_m + \epsilon_m$. Similarly, we optimize $\hat{x}_m$ through $T$ iterations and use $\hat{x}_m^{(t)}$ to denote $\hat{x}_m$ at $t$-th optimization step.

### 3.4 Leveraging Out-of-Distribution Data

Beyond translating majority samples in the ID setting (e.g., samples in $\mathcal{D}_{\text{train}}$) to achieve efficient over-sampling and data augmentation for minority classes, it is more practical and valuable by leveraging OOD data to achieve the balance of a long-tailed dataset. Let $\mathcal{D}_{\text{ood}} = \{x_i\}_{i=1}^{N_o}$ denote the OOD dataset, where $x_i \in \mathbb{R}^d$ denotes $i$-th sample. We assume that the OOD dataset is unlabeled or the label information is not useful due to the large distribution shift from the ID dataset $\mathcal{D}_{\text{train}}$. Now we can initialize the minority sample as $\hat{x}_m := x_{\text{ood},m} + \epsilon_m$, where $x_{\text{ood},m}$ is a randomly sampled image from $\mathcal{D}_{\text{ood}}$ for $\hat{x}_m$.

Recall that M2m restricts the target classifier $f$ to have lower confidence on the original class $k_0$ of $x_0$ by adding a regularization term in Eq. 3. However, the introduction of the OOD dataset brings a challenge as there is no corresponding $k_0$ for each $x_{\text{ood},m}$ in $\mathcal{D}_{\text{ood}}$. Therefore, we replace the constraint of the synthetic samples about $k_0$ by introducing a confusing class $k_c$. Specifically, we obtain the confusion matrix $\mathbf{A} \in \mathbb{R}^{K \times K}$ using a randomly sampled balanced subset from $\mathcal{D}_{\text{train}}$ and the pre-trained classifier $g$, whose element $A_{ij}$ denotes the probability that a sample belongs to class $i$ but is predicted as class $j$. Then, for the target minority class $k$, $k_c$ is its most confusing class if $A_{k,k_c} \geq A_{k,i}$, where $i \in [1, K]$ and $i \neq k$. Finally, we design the constraint on the confusing class for an optimized sample $\hat{x}_m$ as $f_{k_c}(\hat{x}_m)$ and rewrite Eq. 7 as:

$$\hat{x}_m^{(t)} = \underset{\{\hat{x}_m := x_{\text{ood},m}\}_{m=1}^{M_k}}{\arg\min} \sum_{m=1}^{M_k} \left[ \mathcal{L}(g(\hat{x}_m), k) + \lambda_1 f_{k_c}(\hat{x}_m) \right] + \lambda_2 \mathbb{E}_{\theta \sim P_\theta}[\text{Dist}_\theta(P_k, Q_k)], \tag{8}$$

where $\hat{x}_m$ is initialized by an OOD sample $x_{\text{ood}}$ with noise $\epsilon$, and $f_{k_c}(\hat{x}_m)$ restricts $f$ to have lower confidence on the confusing class $k_c$. That is to say, we should avoid the synthetic samples to contain significant information of the confusing class in the viewpoint of target classifier $f$. In addition to addressing the issue of exploiting $\mathcal{D}_{\text{ood}}$, this regularization term can not only address the issue of exploiting OOD but also be added to the training loss in the ID scenario.

### 3.5 Implementation Details

**Mini-batch learning.** We adopt the stochastic gradient descent (SGD) (Ruder, 2016) to learn the target classifier $f$ and optimize the synthetic samples based on a batch-wise re-sampling. More specifically, we use a standard over-sampling (Huang et al., 2016) to obtain a class-balanced mini-batch $\{(x_i, y_i)_{i=1}^B\}$. To stimulate the generation of $N_1 - N_k$ samples for any $k$, for each sample $x_i$ in the mini-batch, we use probability $\frac{N_1 - N_{y_i}}{N_1}$ to decide whether learning a synthetic sample $\hat{x}_i$ to replace $x_i$. Following this, a joint learning step occurs within the mini-batch: we re-sample a randomly initialized $\psi_\theta$ to serve as an embedding function, then optimize the synthetic samples guided by our method, which are immediately used to update the target classifier $f$. We give a whole training paradigm in Alg. 3 in Appendix. B.

---

**Algorithm 1:** Oversampling Minority Samples via Our Method (In-Distribution).

---

**Input** : $\mathcal{D}_{\text{train}}$, classifier $f$, pre-trained classifier $g$ and hyper-parameters.

**1** Initialize $\mathcal{D}_{\text{bal}} \leftarrow \mathcal{D}_{\text{train}}$;

**2 for** $k = 2, ..., K$ **do**

**3**     Compute $M_k \leftarrow N_1 - N_k$;

**4**     Initialize $\hat{\mathcal{D}}_k \leftarrow \emptyset$;

**5**     **for** $m = 1, ..., M_k$ **do**                           `// Step 1.  Sample selection`

**6**        Sample a majority class $k_0$ with $p = 1 - \beta^{(N_{k_0} - N_k)^+}$;

**7**        Sample a $x_m$ from $k_0$ ;

**8**        Initialize $\hat{x}_m^{(0)} \leftarrow x_m + \epsilon_m$ with a Gaussian noise $\epsilon_m$.;

**9**        Update $\hat{\mathcal{D}}_k \leftarrow \hat{\mathcal{D}}_k \cup \{(\hat{x}_m^{(0)}, \hat{y}_m = k)\}$;

**10**     **end**

**11**     Build $Q_k = \sum_{m=1}^{M_k} \frac{1}{M_k} \delta_{\hat{x}_m}$ and $P_k = \sum_{n=1}^{N_k} \frac{1}{N_k} \delta_{x_n}$ according to Eq. 4;

**12**     **for** $t = 1, ..., T$ **do**                                 `// Step 2.  Optimize` $\hat{x}$

**13**        $\hat{x}_m^{(t)} = \underset{\hat{x}_m}{\arg\min} \sum_{m=1}^{M_k} [\mathcal{L}(g(\hat{x}_m), k) + \lambda_1 f_{k_0}(\hat{x}_m)] + \lambda_2 \mathbb{E}_{\theta \sim P_\theta} [\text{Dist}_\theta(P_k, Q_k)]$ according to Eq. 7;

**14**     **end**

**15**     Update $\hat{\mathcal{D}}_k$ with optimized synthetic samples.;

**16**     **for** $\hat{x}_m$ *in* $\hat{\mathcal{D}}_k$ **do**                        `// Step 3.  Sample rejection for` $\hat{x}$

**17**        **if** $\mathcal{L}(g(\hat{x}_m), k) \geq \tau$ *or Reject* $= 1$ **then**

**18**           $\hat{x}_m \leftarrow$ with a random sample from class $k$ in $\mathcal{D}_{\text{train}}$;

**19**        **end**

**20**        Update $\mathcal{D}_{\text{bal}} \leftarrow \mathcal{D}_{\text{bal}} \cup \{(\hat{x}_m, k)\}$;

**21**     **end**

**22 end**

---

**Sample selection criteria for** $x_0$. We choose a seed sample $x_0$ to learn $\hat{x}_i$ for $x_i$ with class $k$. In an OOD setting, we just randomly sample an image from $\mathcal{D}_{\text{ood}}$ as $x_0$. In ID setting, we first choose $k_0$ with the probability $k_0 \sim 1 - \beta^{(N_0 - N_k)^+}$ in the current mini-batch, where $(\cdot)^+ := \max(\cdot, 0)$, and $\beta \in [0, 1)$ is a hyper-parameter. After that, $x_0$ is sampled uniformly among samples in class $k_0$. Once we choose the seed sample $x_0$ for the minority class $k$, we start to learn $\hat{x}$. Rather than using all $N_k$ samples within class $k$, we randomly sample a subset of the real samples from class $k$ to construct $P_k$ in each iteration to reduce computation. Besides, we use the to-be-optimized samples for class $k$ in the current mini-batch to build $Q_k$. Finally, we optimize $\hat{x}$ using Eq. 7 or Eq. 8 by performing $T$ iterations with a step size of $\eta$, depending on ID or OOD, respectively.

**Sample rejection criteria for** $\hat{x}$. To reduce the harmfulness of unreliable synthetic samples, it is necessary to design sample rejection criteria to discard unsatisfactory synthetic samples. Here, we consider two conditions that can determine a reliable synthetic sample. Following M2m, the first one is setting a threshold $\tau > 0$ and rejecting the resultant synthetic sample for $k$-th class if $\mathcal{L}(g; \hat{x}, k) > \tau$ for stability. For the second factor, M2m designs the rejection probability as $\mathbb{P}(\text{Reject } \hat{x} \mid k_0, k) \propto \beta^{(N_{k_0} - N_k)^+}$. Different from M2m that utilizes the class frequency of $k_0$ and target class $k$ to decide the reliability of $\hat{x}$, we introduce a more general sample-level criteria to reject $\hat{x}$ (*i.e.*, Reject $\hat{x} = 1$) if it satisfies:

$$\frac{1}{N_k} \sum_{n=1}^{N_k} d(\psi_\theta(\hat{x}), \psi_\theta(x_n)) > \frac{1}{N_k^2} \sum_{n=1}^{N_k} \sum_{m=1}^{N_k} d(\psi_\theta(x_n), \psi_\theta(x_m)), \tag{9}$$

where $d(\psi_\theta(\hat{x}), \psi_\theta(x_n))$ indicates the distance between $\hat{x}$ and $x_n$ and can be defined by cosine similarity, *i.e.*, $1 - \cos(\psi_\theta(\hat{x}), \psi_\theta(x_n))$. This rejection criterion can avoid the requirement for $N_{k_0}$, which can also be applied to the OOD scenario. The underlying intuition is that the synthetic samples are expected to have a smaller distance from real samples in class $k$ than the intra-class distance. We replace $x_i$ in the current mini-batch by $\hat{x}$ if it satisfies the above two factors. If $\hat{x}$ dose not satisfy the condition 9 (i.e., *Reject* $= 1$) or the pre-trained biased classifier $g$ still produces a high classification confidence (i.e., $\mathcal{L}(g; \hat{x}, k) > \tau$), we will replace $\hat{x}$ with a real minority sample from $\mathcal{D}_{\text{train}}$. We summarize the synthetic process for the ID setting in Algorithm 1.

Table 1: Test top-1 errors (%) of ResNet-32 on CIFAR-LT-10 / CIFAR-LT-100 under different imbalance factors on the ID setting, where †, ‡ and * denote the results from the original paper, our reproduction and MetaSAug (Li et al., 2021), respectively. Results of SMOTE are from Kim et al. (2020). The methods are trained with CE loss unless otherwise stated.

| Method | CIFAR-LT-10 | | | | CIFAR-LT-100 | | | |
|---|---|---|---|---|---|---|---|---|
| Imbalance Factor | 200 | 100 | 50 | 10 | 200 | 100 | 50 | 10 |
| CE Loss* | 34.13 | 29.86 | 25.06 | 13.82 | 65.30 | 61.54 | 55.98 | 44.27 |
| Focal Loss* | 34.71 | 29.62 | 23.29 | 13.34 | 64.38 | 61.59 | 55.68 | 44.22 |
| CB,CE Loss* | 31.23 | 27.32 | 21.87 | 13.10 | 64.44 | 61.23 | 55.21 | 42.43 |
| LDAM-DRW* | 25.26 | 21.88 | 18.73 | 11.63 | 61.55 | 57.11 | 52.03 | 41.22 |
| MetaSAug† | 23.11 | 19.46 | 15.97 | 10.56 | **60.06** | 53.13 | 48.10 | 38.27 |
| RSG‡ | - | 20.04 | 17.2 | - | - | 55.4 | 51.5 | - |
| MBJ‡ | - | 19.0 | 13.4 | 11.2 | - | 54.2 | 47.4 | 39.3 |
| CB-SAFA† | 27.18 | 23.68 | 19.79 | 12.07 | 60.34 | 54.13 | 52.04 | 39.77 |
| CUDA† | - | - | - | - | - | 57.3±0.4 | 52.8±0.4 | 40.4±0.6 |
| CMO‡ | 25.43 | 19.59 | 16.47 | 11.50 | 63.47 | 56.13 | 51.71 | 40.49 |
| OTMix‡ | - | 21.7 | 16.6 | 9.8 | - | 53.6 | 49.3 | 38.4 |
| M2m† | 25.34±0.46‡ | 21.7±0.16 | 18.81±0.76‡ | 12.5±0.15‡ | 63.77±0.33‡ | 57.1±0.16 | 50.48±0.43‡ | 44.8±0.05‡ |
| OURS | **22.85±0.12** | **18.20±0.21** | **12.96±0.11** | **9.34±0.11** | 61.28±0.21 | **52.95±0.18** | **47.02±0.26** | **37.57±0.32** |

# 4 Experiments

In this section, we present experimental results to show the effectiveness of the proposed method. The detailed experiment settings and hyper-parameters are provided in Appendix C.1.

**Datasets.** We evaluate our method on CIFAR-LT-10 / CIFAR-LT-100, ImageNet-LT and Places-LT. We build CIFAR-LT-10 / CIFAR-LT-100 from the standard CIFAR-10/CIFAR-100 datasets (Krizhevsky et al., 2009) with IF $\in \{50, 100, 200\}$ (Kim et al., 2020; Kang et al., 2019; Li et al., 2021). ImageNet-LT is a subset of the ImageNet-2012 dataset (Deng et al., 2009) with 1000 classes and IF $= 1280/5$ (Kim et al., 2020; Ren et al., 2020). Places-LT is a subset from the Places-365 dataset (Zhou et al., 2017) with 365 classes and IF $= 4980/5$ (Cao et al., 2019; Ren et al., 2020).

**Baselines.** We compare with five types of baselines: (1) ***Cross-entropy (CE)***. (2) ***Re-weighting loss***, including Focal loss (Lin et al., 2017), Class-Balanced (CB) loss (Cui et al., 2019), Balanced-Softmax (BS) loss (Ren et al., 2020) and LDAM-DRW loss (Cao et al., 2019). (3) ***Feature based augmentation methods***, including MetaSAug (Li et al., 2021), SAFA (Hong et al., 2022), CUDA (Ahn et al., 2023), RSG (Wang et al., 2021) and MBJ (Liu et al., 2022). (4) ***Minority over-sampling methods***, including SMOTE (Chawla et al., 2002), M2m (Kim et al., 2020), CMO (Park et al., 2022) and OTMix (Gao et al., 2024b). (5) **OOD methods**, i.e., Open-Sampling (Wei et al., 2022).

## 4.1 Experiments on Long-tailed CIFAR

**Results with the ID setting.** Tab. 1 summarizes the average results of our method for three independent runs with standard deviation on CIFAR-LT-10 / CIFAR-LT-100 under different settings. We find that our method outperforms the CE baseline and re-weighting methods by a large margin. Moreover, our method achieves a significant improvement than both feature- and sample- based data augmentation methods, except for IF $= 200$ with CIFAR-LT-100 when compared with MetaSAug. Remarkably, the comparison between ours and the minority sample synthetic method, *i.e.*, M2m, confirms the validity of introducing the distribution matching loss when transferring the majority samples to the minority classes. Besides, we use MMD and ED to implement $\text{Dist}_\theta$ and report results on CIFAR-LT-10 and time complexity in Section 4.8.

**Results with the OOD setting.** To validate whether our proposed method can translate OOD instances, we employ 300,000 random images[1] (Hendrycks et al., 2018) as the OOD dataset $\mathcal{D}_{\text{ood}}$ for CIFAR-LT-10 / CIFAR-LT-100 by following Open-Sampling (Wei et al., 2022). We report the performance in the case of IF $= 100$ and IF $= 50$. As shown in Tab. 2, we find that our method is significantly better than Open-Sampling (OS), which utilizes open-set noisy labels to re-balance the long-tailed training dataset. It is reasonable since we optimize the OOD samples from the view of classification and distribution matching rather than

---

[1]https://github.com/hendrycks/outlier-exposure

Table 2: Test top-1 errors (%) of ResNet-32 with different imbalance factors on the OOD setting.

| Method | CIFAR-LT-10 | |
|---|---|---|
| IF | 100 | 50 |
| OS | 22.38±0.28 | 18.24±0.51 |
| OURS | **20.03±0.17** | **16.39±0.22** |
| IF | 100 | 50 |
| OS | 59.74±0.65 | 55.23±0.25 |
| OURS | **55.09±0.23** | **49.38±0.27** |

Table 3: Test top-1 errors (%) of ResNet-50 on ImageNet-LT (INLT) and ResNet-152 on Places-LT (PLT).

| Method | INLT | PLT | Method | INLT | PLT |
|---|---|---|---|---|---|
| CE | 58.4 | 70.1 | FSA | - | 63.6 |
| Focal Loss | - | 65.4 | MBJ | - | 61.9 |
| LDAM-DRW | 50.2 | - | CMO + RIDE | 43.8 | - |
| BS | 49.0 | 61.3 | OTMix+CE | 48.0 | - |
| RIDE (3 experts) | 45.1 | - | OTMix+BS | 44.4 | - |
| BCL | 44.0 | - | OTMix+RIDE | 42.7 | - |
| M2m + CE | 55.40 | 63.27 | **OURS+CE** | 53.77 | 61.68 |
| Over-Sampling + CE* | 55.34 | 64.27 | **OURS+BS** | **47.31** | **60.37** |
| RSG + LDAM-DRW | - | 60.7 | **OURS+RIDE** | **43.22** | - |
| MisLAS | 47.3 | - | **OURS+BCL** | **41.89** | - |

endowing them with noisy labels without re-labeling OOD samples. Besides, we perform experiments by combining both OOD and ID settings, which produce a better performance than that in the ID setting. The detailed results are provided in the supplementary materials.

**Boosting other methods.** To investigate whether our method can be combined with other long-tailed methods under ID and OOD settings, we consider several classical re-weighting losses, including CB loss (Cui et al., 2019) and BS loss (Ren et al., 2020). As shown in Tab. 4, our method significantly improves the performance of re-weighting methods under the ID setting and performs better than M2m. Under the OOD setting, M2m is not usable, while the performance of Open-Sampling is worse than our method combined with different re-weighting losses. These results indicate the effectiveness and flexibility of our method when combined with other methods under both ID and OOD settings.

Table 4: Test top-1 errors (%) of ResNet-32 on CIFAR-LT-10 dataset under both ID and OOD settings when combined with different re-weighting methods.

| Method | CIFAR-LT-10 (ID) | | | CIFAR-LT-10 (OOD) | | |
|---|---|---|---|---|---|---|
| IF | 200 | 100 | 50 | 200 | 100 | 50 |
| CE | 34.13 | 29.86 | 25.06 | - | - | - |
| + M2m | 25.34 | 21.70 | 18.81 | - | - | - |
| + OS | - | - | - | 28.28 | 22.38 | 18.24 |
| + OURS | 22.85 | 18.20 | 15.96 | 23.43 | 20.03 | 16.39 |
| Δ | ↓ 2.49 | ↓ 3.50 | ↓ 2.15 | ↓ 4.85 | ↓ 2.35 | ↓ 1.85 |
| CB-DRW | 31.23 | 27.32 | 21.87 | - | - | - |
| + M2m | 25.24 | 19.33 | 18.25 | - | - | - |
| + OS | - | - | - | 29.77 | 24.23 | 19.90 |
| + OURS | 21.19 | 18.07 | 16.30 | 22.69 | 20.16 | 16.70 |
| Δ | ↓ 4.95 | ↓ 1.26 | ↓ 1.95 | ↓ 7.08 | ↓ 4.07 | ↓ 3.20 |
| BS | - | 21.97 | 18.37 | - | - | - |
| + M2m | 25.16 | 23.43 | 19.96 | - | - | - |
| + OS | - | - | - | 28.59 | 20.95 | 17.24 |
| + OURS | 20.98 | 16.13 | 14.22 | 23.08 | 19.81 | 17.06 |
| Δ | ↓ 4.18 | ↓ 7.30 | ↓ 5.74 | ↓ 5.51 | ↓ 1.14 | ↓ 0.08 |

## 4.2 Experiments on ImageNet-LT and Places-LT

**Results.** As summarized in Tab. 3, we perform experiments on ImageNet-LT and Places-LT. We can see that our method using CE loss outperforms the vanilla CE, over-sampling and M2m, which indicates the effectiveness of generating the minority samples from the view of the distribution matching. Furthermore, our method can also be combined with other losses, where we take the BS loss as an example and obtain improvements by 1.69% and 0.93% compared with the BS loss on ImageNet-LT and Places-LT, respectively. These results show that our proposed data augmentation method is effective on large-scale complicated long-tailed datasets.

## 4.3 Ablation Study

We conduct a series of ablation studies to analyze the influence of each term on CIFAR-LT-10, begining with a brief recall of the key components: a) $\mathcal{L}(g(\hat{x}_m), k)$ evaluates how strongly a pre-trained classifier $g$ still associates a synthetic sample $\hat{x}$ with its target (e.g., minority) class $k$. b) $f_{k_0}(\hat{x}_m)$ represents a term measuring the confidence of our target classifier $f$ that the synthetic sample $\hat{x}$ correctly aligns with its original majority label $k_0$. c) $f_{k_c}(\hat{x}_m)$ denotes a term reflecting the likelihood, according to our target classifier $f$, that the synthetic sample $\hat{x}$ is misclassified into a known "confusing" class $k_c$. d) $\mathbb{E}_{\theta \sim P_\theta}[\text{Dist}_\theta(P_k, Q_k)]$ signifies our core distribution matching objective, which aims to minimize the discrepancy (e.g., using Optimal Transport distance) between the feature distributions of real and synthetic samples for a given class $k$. For brevity in the subsequent ablation discussion, we will refer to these components as a) $L(k)$, b) $L(k_0)$, c) $L(k_c)$, and d) $L(D)$, respectively.

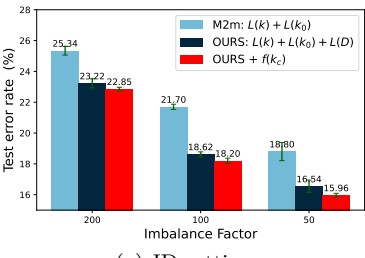 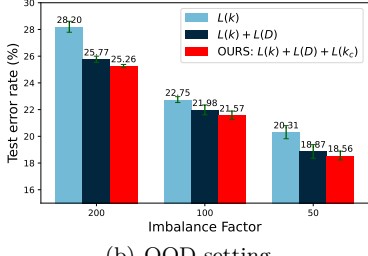 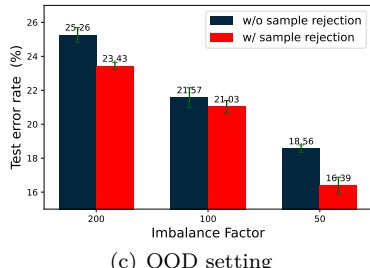

|(a) ID setting|(b) OOD setting|(c) OOD setting|

Figure 1: Ablation studies of our method on CIFAR-LT-10 with varying imbalance factors.

**Distribution Matching Loss ($L(D)$).** To verify the effect of our proposed distribution matching loss, $L(D)$, we first compare M2m (which utilizes an objective based on $L(k) + L(k_0)$) with our method (using $L(k) + L(k_0) + L(D)$) in Fig. 1(a) under the ID setting. It is clear that our method consistently outperforms M2m across different imbalance factors. This improvement stems from $L(D)$ guiding the synthetic sample distribution to closely match that of the real samples, thereby enabling our method to generate more effective synthetic samples. Additionally, M2m's reliance on the $L(k_0)$ term (target class confidence for $\hat{x}_m$) makes it challenging to apply directly to OOD samples where the target label $k_0$ for an OOD input is not predefined. For OOD scenarios, we therefore compare an objective using only $L(k)$ against one that combines $L(k) + L(D)$ for minority sample synthesis. As shown in Fig. 1(b), the latter configuration, incorporating our $L(D)$ term, achieves better performance.

**Confusing Class Regularization ($L(k_c)$).** While M2m is primarily designed for ID settings, our method's applicability can extend to OOD scenarios, partly facilitated by introducing the $L(k_c)$ term. This term addresses potential misclassification of a synthetic sample (intended for a target minority class $k_t$) into a frequently confused class $k_c$. As shown in Fig. 1(b) for OOD settings, our method incorporating an objective of $L(k) + L(k_c) + L(D)$ achieves superior performance compared to an objective with only $L(k) + L(D)$ across various imbalance factors. Notably, this $L(k_c)$ regularization is also beneficial in the ID setting. Fig. 1(a) demonstrates that our method gains further performance improvements in ID scenarios with the inclusion of $L(k_c)$. These results illustrate that the $L(k_c)$ term not only aids OOD applicability but also enhances ID performance, by promoting the generation of higher-quality synthetic minority samples that are more discriminable from their confusing classes.

**Sample rejection in the OOD setting.** As specified in previous section, the unreliable generation quality of synthetic samples urges us to propose the sample rejection criteria, especially in the OOD setting. To validate our proposed rejection strategy Eq. 9, we perform an ablation study on the CIFAR-LT-10 with IF = 10. Results in Fig. 1(c) present that using our proposed rejection strategy consistently improves the performance when leveraging the OOD samples for CIFAR-LT-10 with different imbalance factors. That is to say, our method can fully use OOD samples to generate minority samples while alleviating the toxicity of the distribution shift brought about by OOD samples.

**Embedding spaces** $\mathbb{E}_{\theta \sim P_\theta}[\cdot]$. We investigate the effect of the embedding for computing the distribution matching loss described in Eq. 7. We use the encoder in the to-be-learned model $f$ and that in the pre-trained model $g$ as the baselines, where we also discuss the performance of a randomly initialized encoder and a CLIP vision encoder (Radford et al., 2021). All encoders have the same architecture for a fair comparison, except for CLIP. Besides, we also consider a non-neural network model, Principal Component Analysis (PCA). As summarized in Tab. 5, the PCA serves as the worst embedding

Table 5: Test top-1 errors (%) with different embeddings.

| Feature Extractor | CIFAR-LT-10(ID) | | |
|---|---|---|---|
| | 200 | 100 | 50 |
| PCA | 28.37 | 23.82 | 21.98 |
| Model $g$ | 25.60 | 21.24 | 16.55 |
| Model $f$ | 25.72 | 20.03 | 16.88 |
| A randomly initialized encoder | 25.32 | 19.78 | 16.57 |
| CLIP (RN50) | 23.32 | 18.94 | 16.21 |
| The family of random encoders | **22.85** | **18.20** | **15.96** |

function. The possible reason behind this is the principal components derived from samples can be quite variable and may not always capture the most discriminative features relevant to the overall distribution, especially compared to the features extracted by neural networks. Besides, using the encoder in the imperfect pre-trained $g$ and $f$ also achieves inferior results. It is reasonable since $g$ is a biased model and cannot extract satisfactory features, and the optimization of $f$ is coupled with that of synthetic samples. Interestingly, we

Table 6: Test top-1 errors (%) on CIFAR-LT-10 with IF $\in [100, 50]$ under the ID and OOD settings. For the left part, We evaluate the influence of sample rejection criteria and sample selection criteria, where $\tau = 0.9$ and $\beta = 0.999$. For the right part, we evaluate the influence of different $\tau$ and $\beta$. All the experiments are conducted by using sample selection criteria and sample rejection criteria.

| - | Selection | Rejection | 100 | 50 | - | $\tau$ | $\beta$ | 100 | 50 |
|---|---|---|---|---|---|---|---|---|---|
| ID | $k_0$ | - | 31.16 | 27.25 | ID | 0.9 | 0.999 | 18.20 | 15.96 |
| ID | $k_0$ | $L(p)$ | 20.73 | 20.86 | ID | 0.6 | 0.999 | 18.69 | 16.33 |
| ID | $k_0$ | $L(p) + L(d)$ | 18.20 | 15.96 | ID | 0.3 | 0.999 | 19.12 | 16.82 |
| ID | Random | $L(p) + L(d)$ | 20.07 | 16.32 | ID | 0.9 | 0.888 | 18.53 | 16.27 |
| OOD | Random | $L(p)$ | 20.32 | 16.56 | ID | 0.9 | 0.777 | 18.97 | 16.59 |
| OOD | Random | $L(p) + L(d)$ | 20.03 | 16.39 | ID | 0.9 | 0.666 | 18.77 | 16.30 |

find that only using a randomly initialized encoder during the entire training process can produce acceptable performance, proving the effectiveness of the appropriate and unbiased embedding function for the loss of distribution matching. Moreover, randomly initializing the embedding function at each training iteration outperforms other settings. It shows that the family of embedding spaces can be obtained by sampling randomly initialized DNNs, and is effective in computing the distance between real and synthetic samples. Moreover, we find that even the CLIP (Radford et al., 2021) visual encoder can be used for our distribution matching purposes.

## 4.4 Detailed ablation study on sample selection and rejection criteria

In this section, we present a detailed ablation study on sample selection strategies and two key rejection criteria for synthesized samples, which we denote $L(p)$ and $L(d)$ for brevity in this discussion. The $L(p)$ criterion, is proposed by M2m from a **p**robability perspective, which rejects a synthetic sample $\hat{x}$ if a biased pre-trained classifier $g$ can not confidently classify it as its target minority label $k$ (requiring its confidence score $\mathcal{L}(g; \hat{x}, k) > \tau$). Our proposed $L(d)$ criterion, with its precise mathematical condition detailed in Eq. 9 drawn from a representation **d**istance view, evaluates whether $\hat{x}$ is sufficiently close geometrically to the real minority samples of the target class to be accepted for training. We now examine the impact of applying these criteria.

As shown in the left part of Tab. 6, the model performs worst under the ID setting when using only the $k_0$ selection criterion and no rejection criteria, regardless of IF values being 50 or 100. However, the introduction of the $L(p)$ rejection criterion for synthesized samples results in a marked improvement, with test errors decreasing from 31.16% to 20.73% for IF = 100 and from 27.25% to 20.86% for IF = 50. This finding underscores the efficacy of $L(p)$ in discerning and filtering out less reliable synthesized samples, leading to improved model performance. Furthermore, the subsequent incorporation of our proposed $L(d)$ criterion contributes to an even more pronounced decrease in test errors, underscoring its effectiveness in further refining the selection of high-quality synthetic samples. These conclusions are corroborated by results in the OOD setting, which demonstrate that our combined rejection strategy ($L(p)+L(d)$) effectively identifies high-quality, credible samples for model training, thereby catalyzing performance gains across different distribution settings. Finally, when maintaining $L(p) + L(d)$ as the rejection mechanism and evaluating different sample selection criteria, we observe that the $k_0 \sim 1 - \beta^{(N_0, N_k)^+}$ approach[2] demonstrates superior performance, surpassing the alternative of randomly selecting a seed sample for initialization.

In the right part of Tab. 6, we evaluate the influence of different values for the threshold $\tau_p$ (for the $L(p)$ criterion) and $\beta$ (related to $k_0$ sample selection) on performance, using the combined $L(p) + L(d)$ rejection criteria. While an excessively lenient $\tau_p$ (a very high value) could theoretically allow more lower-quality synthetic samples to pass (those $L(p)$ aims to filter) and potentially lead to suboptimal performance, our experiments reveal a nuanced trend within the tested range. Specifically, with $\beta$ fixed at 0.999, we observe that test error decreases as $\tau_p$ increases from 0.3 to 0.9. This suggests that, within this range and in conjunction with our $L(d)$ criterion effectively filtering samples by their proximity to the target class distribution, a less stringent $L(p)$ (higher $\tau_p$) allows for a beneficial set of samples to be considered. This indicates that our

---

[2]Assuming $N_k$ here refers to a class count relevant to the $k_0$ selection, ensure consistency with its definition elsewhere.

Table 7: Test top-1 errors (%) of ResNet-32 on CIFAR-LT-10 under different imbalance factors, where †and ‡denote the results from the original paper and our reproduction, respectively. The methods are trained with CE loss unless otherwise stated.

| Initialization | Method | CIFAR-LT-10 | | |
|---|---|---|---|---|
| Source | Imbalance Factor | 200 | 100 | 50 |
| ID | M2m | 25.34‡ | 21.7† | 18.81† |
| OOD | Open-Sampling | 28.28‡ | 22.38† | 18.24† |
| OOD | OURS | 23.43 | 20.03 | 16.39 |
| ID | OURS | 22.85 | 18.20 | 15.96 |
| OOD to ID | OURS | 21.83↓ 1.02 | 17.96↓ 0.24 | 15.32↓ 0.64 |

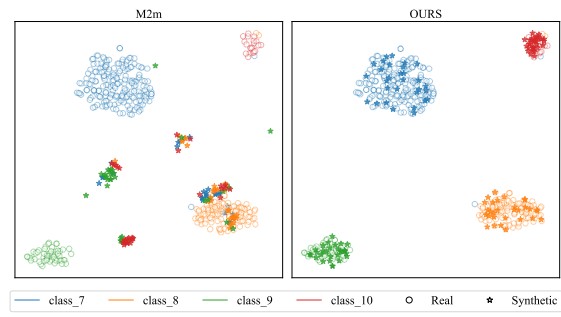

Figure 2: Visualization of features of synthetic/real samples on CIFAR-LT-10 (IF = 100) with ResNet-32, where 'o' and the star indicate real and synthetic samples in the same class, respectively.

overall rejection framework $(L(p) + L(d))$ effectively manages sample quality across different configurations of $\tau_p$. Regarding $\beta$, which influences the diversity of synthetic sample initialization, a larger $\beta$ (when $\tau_p$ is 0.9) yields the best performance, highlighting that greater initialization diversity aids generalization on long-tailed datasets.

## 4.5 Combining OOD and ID.

Beyond leveraging the OOD samples to replace the majority samples in our framework, further, we explore whether combining the OOD setting and ID setting can produce better performance. To this end, we conduct experiments on CIFAR-LT-10 with IF $\in \{200, 100, 50\}$ on ResNet-32. Specifically, we firstly optimize synthetic samples and train the target classifier $f$ using $\mathcal{D}_{ood}$. Then we save the best checkpoint and employ an additional 20 epochs to further train $f$ under the ID setting, where we initialize the minority samples with majority samples, using the Alg. 1. In other words, we first use the OOD setting to train the target classifier $f$ and then further train $f$ under ID setting.

As shown in Tab. 7, our method in ID and OOD settings outperform the M2m and Open-Sampling, respectively. Furthermore, introducing the OOD dataset into the ID setting, our method further achieves 1.02%, 0.24% and 0.64% gains with IF $\in \{200, 100, 50\}$, respectively. These demonstrate that our framework in ID or OOD setting can achieve better performance than corresponding baselines. Besides, the OOD samples can be utilized to further improve the performance of our proposed method under the ID setting.

## 4.6 Visualizations

**Visualization of synthetic samples in feature space.** As shown in Fig. 2, we visualize synthetic minority samples and real minority samples in CIFAR-LT-10 (IF = 100) in feature space, using t-SNE (Van der Maaten & Hinton, 2008). We show classes 7, 8, 9, and 10 (descending ranked by the number of their samples), each of which has 232, 139, 83, and 50 real samples. We randomly select 50 synthetic samples for each class after using the sample rejection criteria. In terms of M2m, synthetic samples from each class are difficult to capture the corresponding real distribution. Besides, synthetic samples from different classes are seriously coupled together. As expected, our synthetic samples can effectively capture the real distribution of each class. Therefore, it reveals why our method can generate more beneficial synthetic minority samples than M2m.

**Visualization of synthetic samples in pixel space.** We use M2m and ours to optimize a sample $x_0$ with $k_0 = $ car to $k = $ deer on the same pre-trained model $g$. Figure 3(a) shows that $x_0$ is correctly classified as a car with a probability of 0.99 on $g$. Then, after optimization, ours and M2m produce different synthetic samples $\hat{x}$ and corresponding noise, even though the two $\hat{x}$ are visually indistinguishable. At this time, the probability of $\hat{x}$ optimized by ours being classified as its original class $k_0$ on $g$ is 0.07, and the probability of being classified to $k$ is 0.91, while the corresponding probabilities of M2m are 0.19 and 0.73. This shows that ours successfully pushes the synthesized sample away from its original label $k_0$ on $g$, and makes it closer to our target label $k$. At the same time, ours also makes $f$ believe that $\hat{x}$ is a sample from $k$ with a higher probability (0.57, 0.32 larger than M2m) on the target classifier $f$, and its classification probability on $k_0$ is

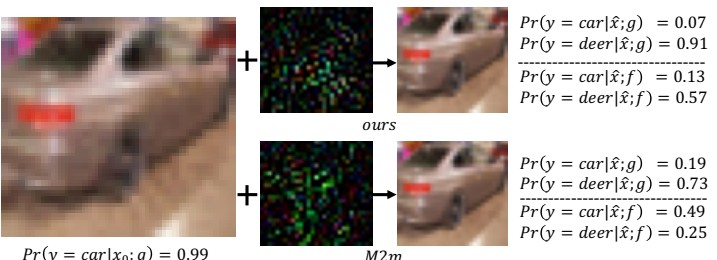

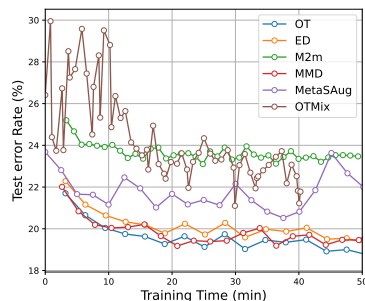

(a) An illustration of a synthetic minority sample by our method and M2m, where g is assumed to be ResNet-32 trained by standard ERM. The noise image is amplified by 20 for better visibility.

(b) Top-1 test errors (%) with the increased training time on CIFAR-LT-10 with IF = 100 trained on ResNet-32.

Figure 3: Visualization of synthetic sample (a) and time-consuming analysis (b).

Table 8: Test top-1 errors (%) of ResNet-32 on CIFAR-LT-10 / CIFAR-LT-100 under different imbalance factors, where †and ‡denote the results from the original paper and our reproduction, respectively. The methods are trained with CE loss unless otherwise stated.

| Method | CIFAR-LT-10 | | | CIFAR-LT-100 | | |
|---|---|---|---|---|---|---|
| Imbalance Factor | 200 | 100 | 50 | 200 | 100 | 50 |
| M2m† | 25.34±0.46‡ | 21.7±0.16 | 18.81±0.76‡ | 63.77±0.33‡ | 57.1±0.16 | 50.48±0.43‡ |
| OURS+MMD | 23.01±0.37 | 18.91±0.29 | 16.42±0.15 | 62.28±0.27 | 53.84±0.15 | 47.25±0.17 |
| OURS+ED | 22.93±0.24 | 18.75±0.15 | 16.16±0.22 | 61.97±0.10 | 53.01±0.22 | 47.33±0.28 |
| OURS+OT | **22.85±0.12** | **18.20±0.21** | **15.96±0.11** | **61.28±0.21** | **52.95±0.18** | **47.02±0.26** |

significantly lower than the corresponding results of M2m by 0.30. This shows that the samples synthesized by our method are more credible and realistic for $f$.

In fact, we do not optimize a majority sample $x_0$ into a picture that is similar to a minority sample in the pixel space, like a generator. Instead, we simply optimize the sample $x_0$ directly to confuse our models $g$ and $f$, making the models believe that our synthetic samples $\hat{x}$ are indeed from real minority class, which helps the network to generalize on the minority class. Recall the visualization of feature space, which shows that the feature distributions of our synthetic samples $\hat{x}$ are closer to real samples $x$ of class $k$. This indicates that the features of our synthetic samples are more realistic and credible, where $f$ regards the features of the real samples and the synthetic samples are from the same distribution. On the other hand, Section C.6 shows that the probability of our synthetic samples being correctly classified as the target class $k$ on the target classifier $f$ is significantly higher than that of M2m. From the classification perspective, our method can produce better synthetic minority samples and reduce the difficulty of the model learning.

In summary, our starting point is to generate synthetic samples that are more realistic and credible for the model. From the perspective of pixel space, our method and M2m have no obvious difference. However, from the classification probability and feature matching perspectives, our synthetic samples are more realistic and credible for the network and, therefore, more effective than M2m.

## 4.7 Comparison of different implements of $\text{Dist}_\theta(P_k, Q_k)$

To prove the generality and effectiveness of our method, we conduct experiments on CIFAR-LT-10 and CIFAR-LT-100 with different imbalanced factors (IF) by using different implements of $\text{Dist}_\theta(P_k, Q_k)$, where the results of different methods are summarized in Tab. 8. We implement $\text{Dist}_\theta(P_k, Q_k)$ using MMD, ED and OT, denoted as OURS+MMD, OURS+ED and OURS+OT, respectively. We report the average results of our method for three runs with the standard deviation independently. We can find that all of them have superior performance compared to the M2m baseline by a large margin. Besides, we can observe that ours+OT performs best, which might benefit the more accurate characterization and measurement of the distance between distributions brought by OT. In other words, OT learns an optimal transport plan which endows each cost element with corresponding importance $T_{ij}$, demonstrating the effectiveness and generality of our proposed method.

### 4.8 Convergence and time complexity

To fairly compare training time-consuming, we conduct an experiment on CIFAR-LT-10 with IF = 100 on ResNet-32 in the same device environment with one Tesla-V100 GPU and we run all the methods with the same epochs. As shown in Fig.3(b), our method (OT, ED and MMD) achieves a lower test error rate within the same computation time, regardless of whether Optimal Transport (OT), Maximum Mean Discrepancy (MMD), or Euclidean Distance (ED) is used to calculate $\text{Dist}_\theta(P_k, Q_k)$.

### 4.9 Additional Analysis

We analyze the impact of OOD initialization source on synthesis in Appendix C.5, classification confidence of synthetic samples on the target classifier $f$ in Appendix C.6, influence of the pre-trained model $g$ in Appendix C.8, visualization of the confusion matrix in Appendix C.9, and more visualization of synthetic samples in Appendix C.10.

## 5 Related Work

**Over-sampling methods for long-tailed problem.** Data-based methods aim to solve the imbalance problem by building relatively balanced classes from the perspective of data, including under-sampling majority samples (He & Garcia, 2009; Drummond & Holte, 2003), over-sampling minority samples (Shen et al., 2016; Buda et al., 2018; Barandela et al., 2004) and data augmentation (Ahn et al., 2023; Park et al., 2022; Yan et al., 2019; Kim et al., 2020; Gao et al., 2023; 2024a; Li et al., 2025; Guo et al., 2022b). Our method has a close connection with minority over-sampling methods. A related work is Optimal transport over-sampling (OTOS) (Yan et al., 2019), which maps the noise to synthetic ones based on the Wasserstein barycenter. Different from OTOS which generates samples by a mapping matrix and is limited to a binary classification, we provide a more general and direct optimization objective for generating synthetic samples. By minimizing this objective, we can obtain synthetic samples with reliable classification confidence and high representation similarity, where we can handle multi-class classification task and leverage more practical OOD setting. Another related work, M2m (Kim et al., 2020), translates majority samples to the target minority class by maximizing the prediction probability. However, in our work, we optimize synthetic minority samples from both perspectives of classification confidence and distribution matching, where we extend the ID to OOD setting for further versatility.

**Utilizing auxiliary dataset for long-tailed problem.** In imbalanced learning, Yang & Xu (2020) leverage unlabeled ID data as additional samples to compensate for the minority classes, while Su et al. (2021) adopts a semi-supervised learning framework to incorporate out-of-class samples from related classes. Open-Sampling (Wei et al., 2022) explores the benefit of using OOD data in the long-tailed problem. The major difference between ours and Open-Sampling is that we translate OOD samples by introducing an optimization phase and introducing a sample rejection strategy but Open-Sampling assigns a noisy label to each OOD sample using a pre-defined label distribution without filtering the OOD data.

## 6 Conclusion

To address the long-tailed classification issue, we propose a novel framework for translating majority samples into synthetic minority samples by leveraging classification confidence and distribution matching. Our method optimizes the synthetic minority samples by enforcing them to satisfy the classification constraints and being close to the distribution of real samples in the target minority class. In addition, we introduce an effective regularization term for confusing classes, enabling our framework to better utilize available and rich OOD data to synthesize minority classes. Extensive experiments on benchmark datasets demonstrate that our framework can generate effective minority samples and achieve the desired long-tailed classification performance.

## Acknowledgment

This work of Dandan Guo was supported by the National Natural Science Foundation of China (NSFC) under Grant 62306125. This work of Hongyuan Zha was supported by the National Natural Science Foundation of China (NSFC) under Grant 72495131. This work was supported by the Guangxi Key R&D Project (No. AB24010167), the Project (No. 20232ABC03A25), Shenzhen Longgang District Science and Technology Innovation Special Fund (No. LGKCYLWS2023018), Futian Healthcare Research Project (No.FTWS002), and Shenzhen Medical Research Fund (No. C2401036), and Hospital University United Fund of The Second Affiliated Hospital, School of Medicine, The Chinese University of Hong Kong, Shenzhen (No. HUUF-MS-202303).

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

# Appendix For
# Synthesizing Minority Samples for Long-tailed Classification via Distribution Matching

## A  Alternatives for $\mathbf{Dist}_\theta(P_k, Q_k)$

### A.1  Maximum Mean Discrepancy (MMD)

**Preliminaries.** MMD is an effective non-parametric metric for comparing the distributions based on two sets of data (Gretton et al., 2012), where the general MMD between two distributions $P$ and $Q$ is defined as

$$\mathrm{MMD}^2(P, Q) = \sup_{\|\phi\|_{\mathcal{H}} \leq 1} \|\mathbb{E}_{x \sim P}\left[\phi\left(x\right)\right] - \mathbb{E}_{y \sim Q}\left[\phi\left(y\right)\right]\|_{\mathcal{H}}^2, \tag{10}$$

where $\mathbb{E}_{x \sim P}[\cdot]$ denotes the expectation with regard to the distribution $P$, $\phi$ is the embedding function, and $\|\phi\|_{\mathcal{H}} \leq 1$ defines a set of functions in the unit ball of a reproducing kernel Hilbert space (RKHS) $\mathcal{H}$.

**Define $\mathbf{Dist}_\theta(P_k, Q_k)$ with MMD.** As we do not have access to ground-truth data distributions for synthetic and real samples shown in Eq. 10, we can use a biased empirical estimate of the MMD by replacing the population expectations with empirical expectations (Gretton et al., 2012), which are computed on the synthetic and real samples in $P_k$ and $Q_k$ and denoted as

$$\mathrm{Dist}_\theta(P_k, Q_k) = \left\| \frac{1}{N_k} \sum_{n=1}^{N_k} \psi_\theta\left(x_n\right) - \frac{1}{M_k} \sum_{m=1}^{M_k} \psi_\theta\left(\hat{x}_m\right) \right\|^2 \tag{11}$$

### A.2  Energy Distance (ED)

**Preliminaries.** Drawing inspiration from the concept of potential energy between objects in a gravitational field, Energy Distance (ED) (Rizzo & Székely, 2016) measures the similarity between two probability distributions, $P$ and $Q$. This can be mathematically expressed as follows:

$$\mathrm{ED}^2(P, Q) = 2\mathbb{E}_{x \sim P, y \sim Q}\|\phi(x) - \phi(y)\| - \mathbb{E}_{x \sim P}\|\phi(x) - \phi(x')\| - \mathbb{E}_{y \sim Q}\|\phi(y) - \phi(y')\|, \tag{12}$$

where $\mathbb{E}_{x \sim P}[\cdot]$ denotes the expectation with respect to the distribution $P$ and $\|\cdot\|$ denotes the Euclidean norm (length) of its argument. In addition, $x'$ and $y'$ are independent copies of $x$ and $y$, respectively.

**Define $\mathbf{Dist}_\theta(P_k, Q_k)$ with ED.** Here, we can define $\mathrm{Dist}_\theta(P_k, Q_k)$ based on the energy distance (ED) as follows:

$$\begin{aligned}
\mathrm{Dist}_\theta(P_k, Q_k) = {} & \frac{2}{N_k M_k} \sum_{n=1}^{N_k} \sum_{m=1}^{M_k} \|\psi_\theta(x_n) - \psi_\theta(\hat{x}_m)\|^2 \\
& - \frac{1}{N_k{}^2} \sum_{n,m=1}^{N_k} \|\psi_\theta(x_n) - \psi_\theta(x_m)\|^2 \\
& - \frac{1}{M_k{}^2} \sum_{n,m=1}^{M_k} \|\psi_\theta(\hat{x}_n) - \psi_\theta(\hat{x}_m)\|^2
\end{aligned} \tag{13}$$

## B  Algorithms of our framework

In this section, we give the algorithm processes 2 of our method under OOD settings as shown in Alg. 2. Our jointly learning framework is shown in Alg. 3.

---

**Algorithm 2:** Oversampling minority samples via our framework (Out-of-Distribution).

---

**Input** : Dataset $\mathcal{D}_{\text{train}}$ and $\mathcal{D}_{\text{ood}}$, classifier $f$ and pre-trained classifier $g$, a confusion matrix $\mathbf{A}$, hyper-parameters:$\{\lambda_1, \lambda_2, \gamma, \eta, T, \tau, \beta\}$

**Output:** A class-balanced dataset $\mathcal{D}_{\text{bal}}$

**1** Initialize $\mathcal{D}_{\text{bal}} \leftarrow \mathcal{D}_{\text{train}}$;

**2** Randomly sample a balanced subset from $\mathcal{D}_{\text{train}}$ and obtain confusion matrix $A$ by evaluate the pre-trained model $g$ using this subset;

**3** **for** $k = 2, ..., K$ **do**

**4**     Compute $M_k \leftarrow N_1 - N_k$; Initialize $\hat{\mathcal{D}}_k \leftarrow \emptyset$;

**5**     **for** $m = 1, ..., M_k$ **do**             // Step 1. Sample selection for $x_0$

**6**        Sample a $x_m$ from $\mathcal{D}_{\text{ood}}$ randomly;

**7**        Initialize $\hat{x}_m^{(0)} \leftarrow x_m + \epsilon_m$ with a standard Gaussian noise $\epsilon_m$.;

**8**        update $\hat{\mathcal{D}}_k \leftarrow \hat{\mathcal{D}}_k \cup \{(\hat{x}_m^{(0)}, k)\}$;

**9**     **end**

**10**     Build $Q_k = \sum_{m=1}^{M_k} \frac{1}{M_k} \delta_{\hat{x}_m} +$ and $P_k = \sum_{n=1}^{N_k} \frac{1}{N_k} \delta_{x_n}$ according to Eq.4. ;

**11**     Obtain $k_c$ if $A_{k,k_c} \geq A_{k,i}$ where $i \in [1, K]$ and $i \neq k$ ;

**12**     **for** $t = 1, ..., T$ **do**               // Step 2. Optimization for $\hat{x}$

**13**        $\hat{x}_m^{(t)} = \sum_{m=1}^{M_k} [\mathcal{L}(g(\hat{x}_m), k) + \lambda_1 \cdot f_{k_c}(\hat{x}_m)] + \lambda_2 \cdot \text{Dist}_\theta(P_k, Q_k)$, according to Eq. 8.

**14**     **end**

**15**     Update $\hat{\mathcal{D}}_k$ with optimized synthetic samples;

**16**     **for** $\hat{x}_m$ *in* $\hat{\mathcal{D}}_k$ **do**           // Step 3. Sample rejection for $\hat{x}$

**17**        **if** $\mathcal{L}(g(\hat{x}_m), k) \geq \tau$ *or Reject* $= 1$ **then**

**18**           $\hat{x}_m \leftarrow$ with a random sample from class $k$ in $\mathcal{D}_{\text{train}}$;

**19**        **end**

**20**        Update $\mathcal{D}_{\text{bal}} \leftarrow \mathcal{D}_{\text{bal}} \cup \{(\hat{x}_m, k)\}$;

**21**     **end**

**22** **end**

---

**Algorithm 3:** Joint Training Paradigm with Synthetic Sample Generation

---

**Input:** Imbalanced dataset $\mathcal{D}$, warm-up epochs $E_{\text{warm}}$, main epochs $E_{\text{main}}$

**Output:** Trained classifier $f$

**1** Randomly initialize classifier $f$;

**2** **for** $e = 1, ..., E_{\text{warm}}$ **do**                  /* Stage 1: Warm-up */

**3**     **for** *mini-batch B in* $\mathcal{D}$ **do**

**4**        Update $f$ using $B$;                  // standard CE training

**5**     **end**

**6** **end**

**7** **for** $e = E_{\text{warm}} + 1, ..., E_{\text{main}}$ **do**         /* Stage 2: Joint training (ours) */

**8**     **for** *mini-batch B in* $\mathcal{D}$ **do**

**9**        Resample an embedding function $\psi_\theta$

**10**        Generate balanced batch $B_{\text{bal}}$ by synthesizing minority samples;     // See Alg. 1

**11**        Update $f$ using $B_{\text{bal}}$;             // train on augmented data

**12**     **end**

**13** **end**

---

# C  More details about datasets and experiments

## C.1  Settings and Training details

Unless otherwise stated, we set the imbalance factor as IF $= N_1/N_K$ and use $T = 5$ iterations with a step size of $\eta = 0.1$ to optimize the synthetic samples at each training iteration. The hyper-parameter for the OT entropy constraint is $\gamma = 0.1$ and the maximum iteration number in the Sinkhorn algorithm is 200. We use SGD with momentum 0.9 and weight decay $5e^{-4}$ and conduct all the experiments on 8 Tesla-V100 GPUs.

## C.2  CIFAR-10 and CIFAR-100 datasets

**CIFAR-LT-10 / CIFAR-LT-100.** The original CIFAR-10/CIFAR-100 datasets (Krizhevsky et al., 2009) include 60,000 images and 10/100 classes with a size of $32 \times 32$, where there are 50,000 images for training and 10,000 for testing. By following (Kim et al., 2020), we create CIFAR-LT-10 and CIFAR-LT-100 by randomly under-sampling in the original datasets with IF $= \{200, 100, 50\}$. We use the original test dataset to evaluate our method. **Training details.** Following (Kim et al., 2020; Li et al., 2021; Guo et al., 2022a), we use ResNet-32 (He et al., 2016) as the backbone. We employ 200 epochs for training $f$ with an initial learning rate $\alpha$ of 0.1, which is decayed by $1e^{-2}$ at 160-th epoch and 180-th epoch. We set batch size as 32 and start our method at 160-th epoch, where we set $\lambda_1$ and $\lambda_2$ as 0.5, $\beta$ as 0.999 and $\tau$ as 0.9.

## C.3  ImageNet-LT and Places-LT

**ImageNet-LT.** The original ImageNet-2012 dataset (Deng et al., 2009) includes 1,281,167 images and 1000 classes with a max size of $1300 \times 732$. By following (Kim et al., 2020; Li et al., 2021; Liu et al., 2019), we create ImageNet-LT with 115.8K samples in 1000 classes and IF $= 1280/5$. We adopt the original validation dataset to test our method.

**Places-LT** The original Places-365 dataset(Zhou et al., 2017) includes 1,803,460 images and 365 classes with a max size of $5000 \times 3068$. By following (Liu et al., 2019), we create Places-LT with 62.5K samples in 1000 classes and the imbalance factor IF $= 4980/5$. We adopt the original test dataset to test our method. **Training details.** Following previous works (Kim et al., 2020; Li et al., 2021; Kang et al., 2019), we use ResNet-50 as the backbone for ImageNet-LT. We employ 200 epochs for training $f$ with an initial learning rate $\alpha$ as 0.1, which will be decayed by $1e^{-1}$ at the 160-th epoch and 180-th epoch. For Places-LT, we employ ResNet-152 pre-trained on the full ImageNet dataset (Russakovsky et al., 2015) as the backbone following (Guo et al., 2022a; Li et al., 2021). We set 200 epochs for training $f$ with an initial learning rate $\alpha$ as 0.1, decayed by $1e^{-1}$ every 40 epochs. We start our method at 160-th epoch for ImageNet-LT and 90-th for Places-LT. We set $\lambda_1$ and $\lambda_2$ as 0.5, $\beta$ as 0.999 and $\tau$ as 0.3. For all experiments, we initialize batch size as 64 and set it as 32 after deploying our method for training stability.

## C.4  Training details about pre-trained model $g$

**CIFAR-LT-10 / CIFAR-LT-100.** For CIFAR-LT-10 / CIFAR-LT-100, we use ResNet-32 (He et al., 2016) as backbone network for pre-training. We employ 200 epochs for training $g$ with an initial learning rate $\alpha$ of 0.1, which will be decayed by $1e^{-2}$ at 160th epoch and 180th epoch. We use SGD with momentum 0.9 and weight decay $5e^{-4}$ and set batch size as 128. In the first 160 epochs, we use the original imbalanced dataset to train the model $g$. For the last 40 epochs, we use the vanilla over-sample technique by inverse class frequency to further train the model $g$. We save the best checkpoint as our pre-trained model $g$.

**ImageNet-LT & Places-LT** For ImageNet-LT, we use ResNet-50 (He et al., 2016) as backbone network for pre-training. We employ 200 epochs for training $g$ with an initial learning rate $\alpha$ of 0.1, which will be decayed by $1e^{-2}$ at 160th epoch and 180th epoch. For Places-LT, we employ ResNet-152 pre-trained on the full ImageNet dataset. We use 120 epochs for training $g$ with an initial learning rate $\alpha$ as 0.1, which is decayed by $1e^{-1}$ every 10 epochs. For both datasets, we use SGD with momentum 0.9 and weight decay $5e^{-4}$ and set batch size as 512. Similar to CIFAR-LT-10 / CIFAR-LT-100, before the $160 - th$ and $90 - th$ epoch on ImageNet-LT and Places-LT, we use the original imbalanced dataset to train model $g$. For the last epochs, we adopt the vanilla over-sample technique by inverse class frequency (Drummond & Holte, 2003) to further train $g$. We save the best checkpoint as our pre-trained model $g$.

## C.5  Impact of OOD Initialization Source on Synthesis

To assess the influence of the initial seed on synthetic sample generation, we evaluated several Out-of-Distribution (OOD) data sources for initialization when targeting the CIFAR-LT-10 dataset, with results presented in Tab. 9. We compared initializations using: (i) Medical images (OrganAMNIST), (ii) Pure

Noise, and (iii) OOD Natural images[3], against a CE-Baseline without synthetic samples. The results clearly demonstrate that our synthesis framework, irrespective of the OOD initialization source, yields substantial improvements over the CE-Baseline across all tested imbalance factors. This underscores the overall benefit of incorporating synthetic samples.

Among the different OOD initialization strategies, a clear performance hierarchy emerged. While initializing with Medical images improved upon the baseline, it was generally outperformed by Pure Noise initialization. This suggests that a very large domain gap (as between medical images and CIFAR-LT-10's natural scenes) can limit the effectiveness of the initial seed. Notably, initializing with OOD Natural images consistently produced the best results, surpassing both Pure Noise and Medical image initializations across all imbalance factors.

These findings highlight that the choice of OOD initialization source is critical, with closer domain proximity to the target dataset leading to superior synthesis outcomes. For natural image classification tasks like CIFAR-LT-10, utilizing OOD natural images as an initialization source is most effective. We attribute this to their ability to provide more relevant foundational features, such as common textures and color distributions, which are advantageous for generating high-quality minority class samples tailored to the target domain.

Table 9: Test top-1 errors (%) of ResNet-32 on CIFAR-LT-10 with IF $\in [200, 100, 50]$ and different OOD datasets. *Distribution* means the initialization of to-be-learned synthetic samples $\hat{x}$, e.g., *OOD* denotes we initialize $\hat{x}$ with OOD samples. *Domain* indicates the corresponding domain of OOD dataset.

| Domain | 200 | 100 | 50 |
|---|---|---|---|
| CE-Baseline w/o synthetic samples | 34.13 | 29.86 | 25.06 |
| Pure Noise | 23.93 | 20.40 | 16.75 |
| Medical | 23.95 | 20.95 | 17.17 |
| Natural | **23.43** | **20.03** | **16.39** |

### C.6 Classification confidence of synthetic sample on target classifier $f$.

To prove the effectiveness of our method in enhancing the generation of high-quality synthetic samples, we compare the classification performance of the synthetic samples on the target classifier $f$ for CIFAR-LT-10 (ID) with IF = 100 on ResNet-32. Specifically, we use the target classifier to output the probability of the synthetic samples in class $k$ being correctly predicted as the $k$ class, where we consider $k \in \{2, \cdots, 10\}$. Then we compute the average probability for all samples and express it as a percentage. Compared to M2m, Fig. 4(a) illustrates that the synthetic samples generated by our method (based on OT) have higher classification confidence for the corresponding concerned class during the training of target classifier $f$, in both ID and OOD settings. This finding suggests that the generated samples by ours are more credible.

### C.7 Proportion of Successfully Synthesized Samples

To quantify the contribution of our novel sample generation relative to the fallback mechanism (repeating existing minority samples), Figure 4(b) illustrates the proportion of successfully synthesized and accepted minority samples within the total augmented minority slots per epoch on CIFAR-10-LT (ID setting). The results show that a consistent fraction, typically ranging from 15% to 21% depending on the imbalance factors, consists of newly generated synthetic samples that pass our quality criteria. This demonstrates that while the fallback ensures robust minority representation, our synthesis module actively produces a continuous stream of novel data, and the overall performance improvement is attributable to both this baseline rebalancing effect and the diversity introduced by these genuinely new instances, distinguishing our approach from simple upsampling.

---

[3]https://github.com/hendrycks/outlier-exposure.

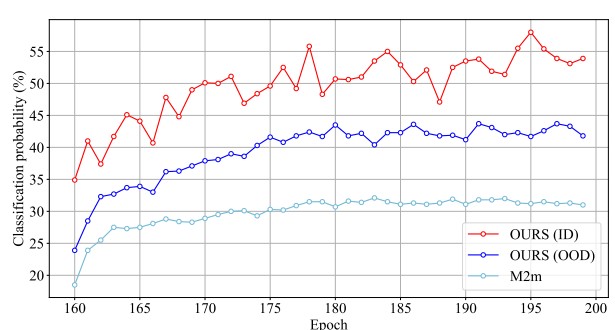
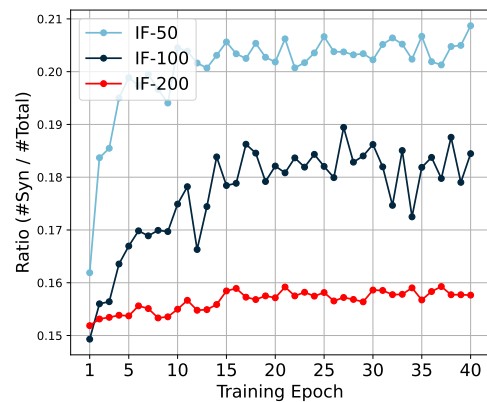

(a) Probability of synthetic samples correctly classified as the concerned class $k$ by the target classifier $f$.

(b) Evolution of synthetic sample ratio under the ID setting across epochs.

Figure 4: Visualization of classification accuracy and out-of-distribution (OOD) sample synthesis behavior.

## C.8 Influence of pre-trained model $g$

In this section, we investigate whether the effectiveness of the proposed method is affected by the performance of pre-trained $g$. We can see that the better the performance of the pre-trained classifier, the better the final classification result of ours in general. Although the pre-trained classifier in the 100th epoch and the best-performed classifier have different classification performance, they have similar impact on the final classification result. Besides, the ensemble of three best-performed classifiers achieves better performance than only using one best-performed classifier. In our work, we perform the experiments only using one best-performed classifier. It demonstrates that the final classification results will increase if we use the ensemble of the pre-trained classifiers to optimize the synthetic samples.

Table 10: Test top-1 errors (%) on CIFAR-LT-10 with IF = 100 under the in-distribution setting. *Pre-trained performance* indicates the overall performance on the $g$ and *Final performance* is the corresponding final classification result.

| $g$ | Pre-trained performance | Final performance |
|---|---|---|
| 1-th epoch | 66.51 | 21.16 |
| 3-th epoch | 59.75 | 20.23 |
| 10-th epoch | 44.26 | 20.17 |
| 20-th epoch | 41.71 | 19.07 |
| 100-th epoch | 30.95 | 18.33 |
| Best epoch | 28.17 | 18.37 |
| Ensemble | 28.77, 30.23, 29.01 | **18.01** |

## C.9 Visualization of confusion matrix

To demonstrate the effectiveness of our method in improving the performance of minority classes, we visualize the confusion matrices of CE, M2m and OURS on CIFAR-LT-10 with IF=200. As shown in Fig.5, CE has poor classification performance in minority classes. Therefore, it is necessary to solve the long-tailed problem. Although M2m mitigates the problem, it still performs poorly on the rarest classes. Our proposed method achieves better performance than CE and M2m. In particular, ours is superior to strong baseline M2m for almost every class, thereby alleviating the imbalanced classification problem.

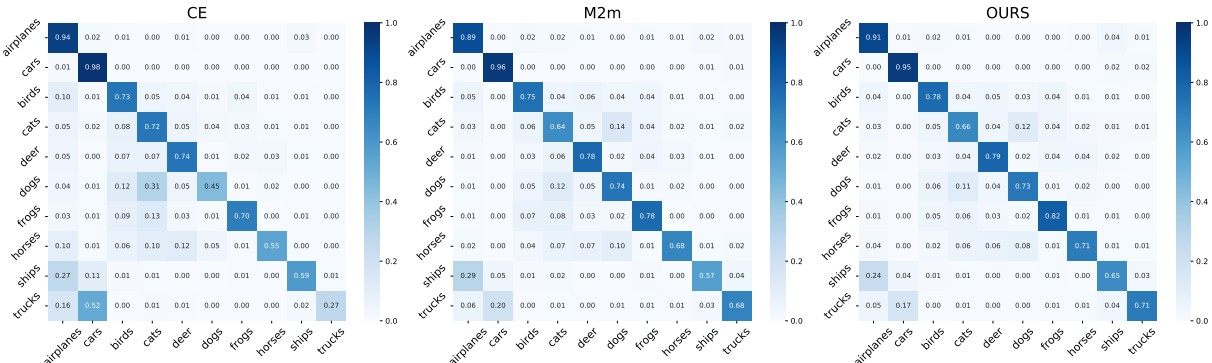

Figure 5: Confusion matrices of the CE, M2m and OURS on CIFAR-LT-10 with the imbalance factor 200.

## C.10  Visualization of Synthetic Samples

In this section, as shown in Fig. 6, Fig. 7, and Fig. 8, we present visual examples of synthetic minority samples generated by our method on the ImageNet-LT dataset. Similarly, while our synthetic samples and their source images may be difficult to distinguish by eye in pixel space, the subtle, optimized "noise" introduced during our synthesis process effectively alters the classifier's perception and classification of these modified samples.

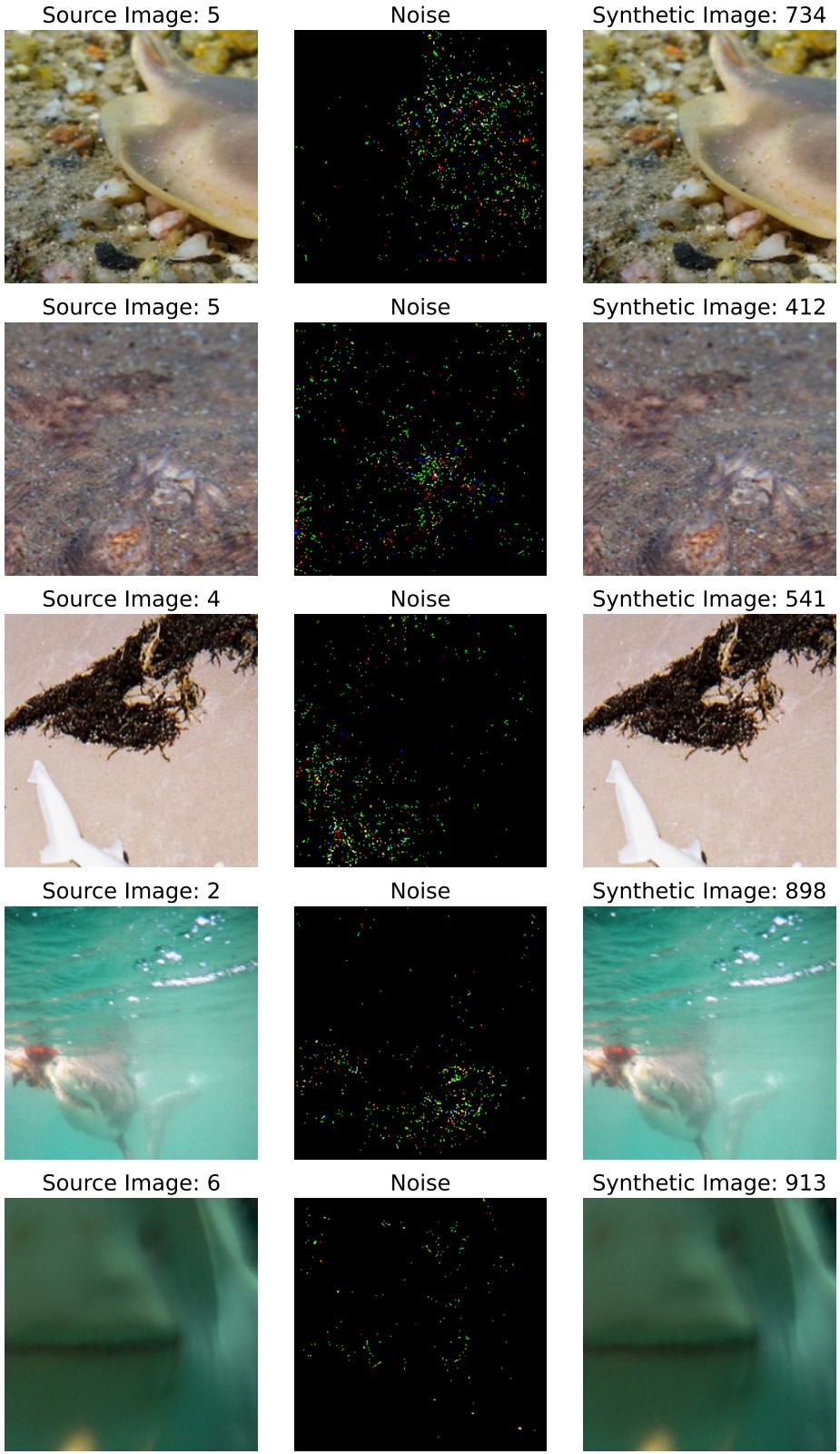

Figure 6: An illustration of synthetic minority samples by our method on ImageNet-LT.

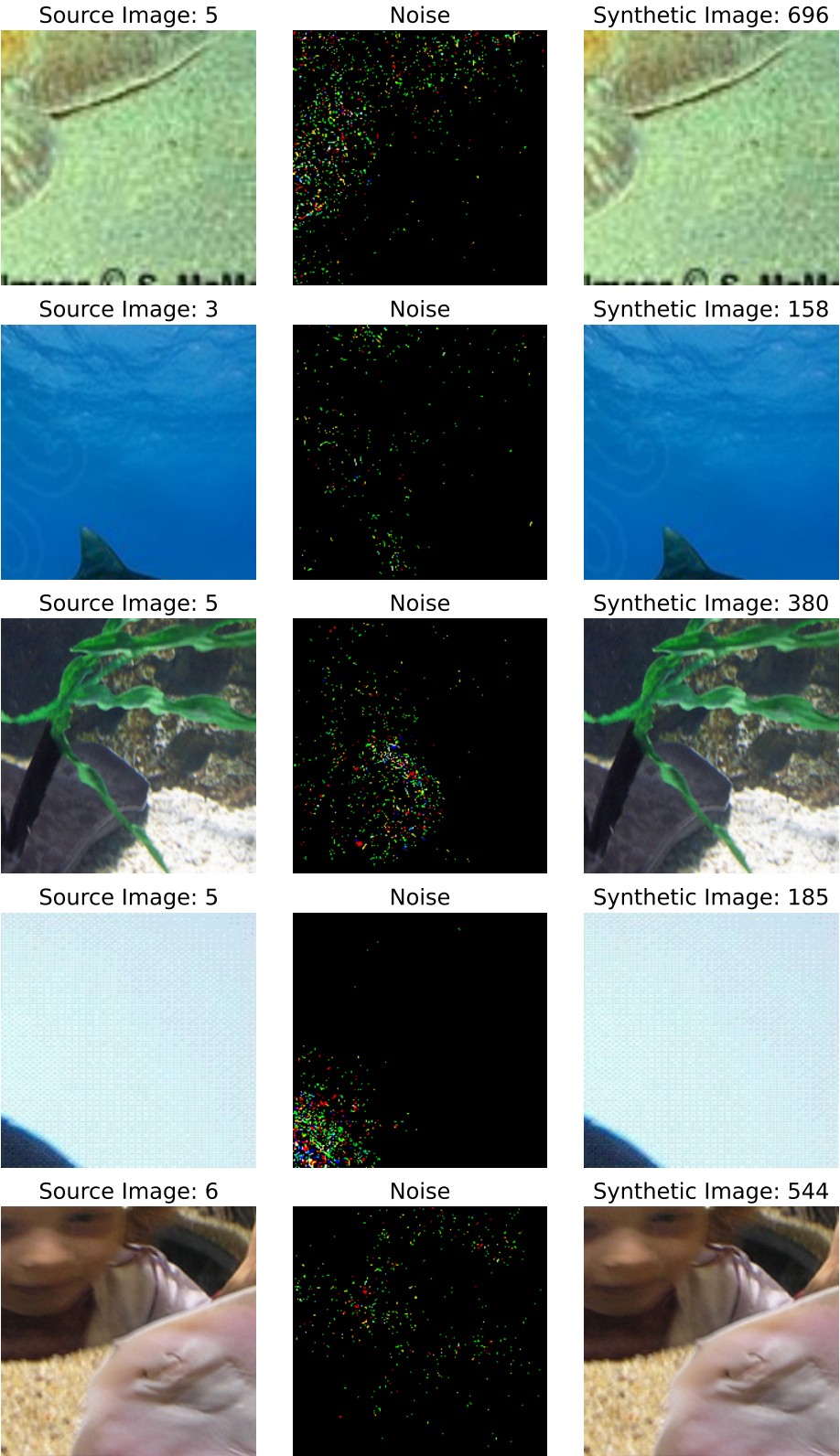

Figure 7: An illustration of synthetic minority samples by our method on ImageNet-LT.

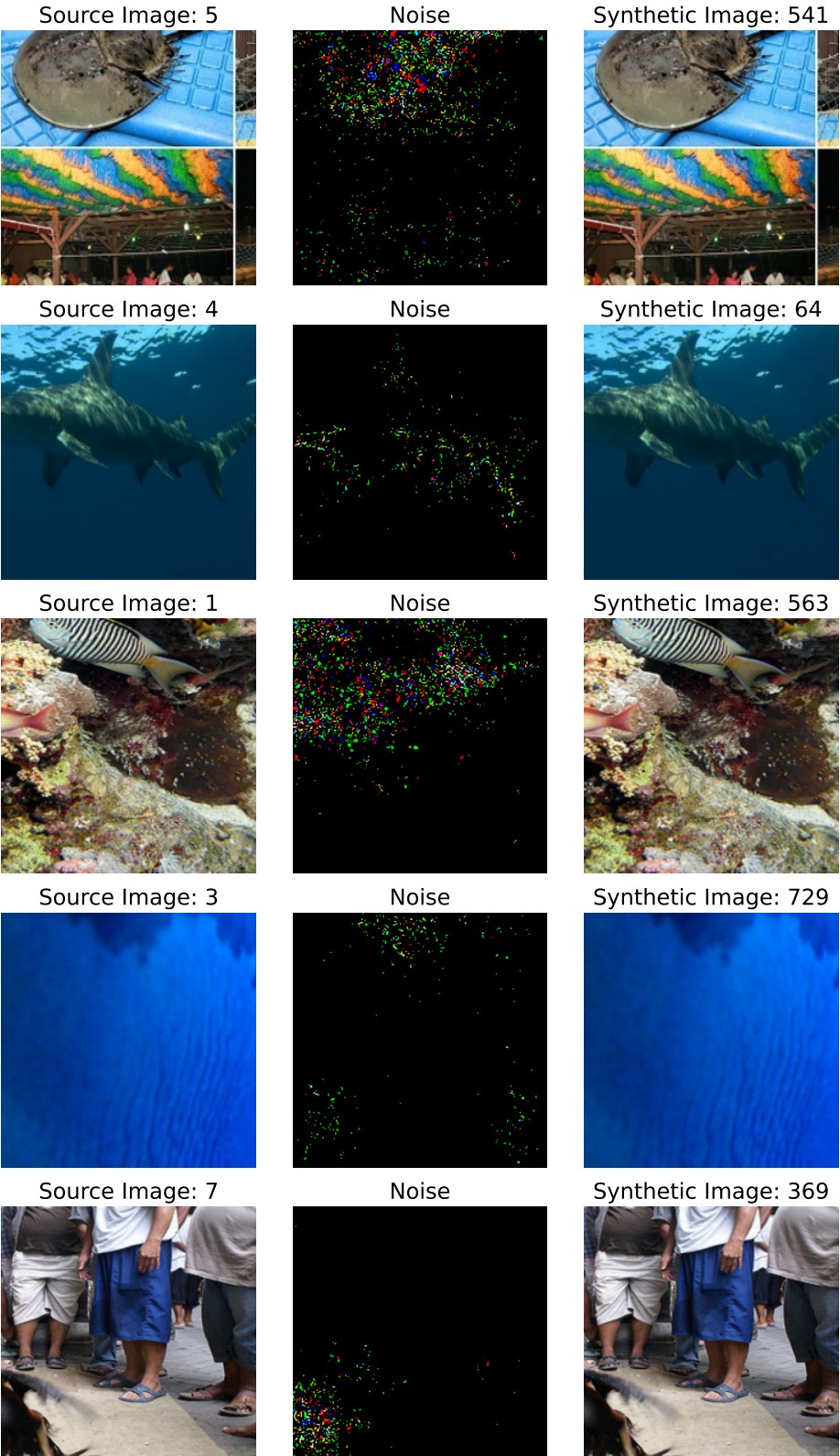

Figure 8: An illustration of synthetic minority samples by our method on ImageNet-LT.

