# OpenReview forum: "Synthesizing Minority Samples for Long-tailed Classification via Distribution Matching"
_TMLR — Accepted by TMLR_

### Review · Reviewer_dhP2 · 2025-04-15

**Summary Of Contributions:**

This paper proposes a novel oversampling framework for long-tailed classification by synthesizing minority class samples via distribution matching. It additionally enforces the synthetic samples to align with the true distribution of the target class using optimal transport. TExtensive experiments on CIFAR-LT, ImageNet-LT, and Places-LT demonstrate improved performance over existing methods.

**Audience:**

Yes

**Claims And Evidence:**

Yes

**Requested Changes:**

see weaknesses

**Strengths And Weaknesses:**

**strengths**
1. The paper provides a solid justification for adding distribution matching.
2. The framework effectively generalizes to OOD settings.
3. The method is extensively tested on multiple benchmarks.

**weakness**
1. The proposed method initializes synthetic samples from majority class examples, but does not explore or justify why this is better than using purely random initialization (e.g., Gaussian noise), which is widely adopted in generative models such as GANs, diffusion models, and DeepDream.
2. The initialization of synthetic samples from majority class examples is insufficiently justified. This is especially problematic when class-wise semantic gaps are large.
3. The expression makes it appear that $\hat{x}$ is deterministic or fixed, which is misleading since $\hat{x}$ is iteratively optimized.
4. It is unclear whether $f$ is jointly updated with synthetic samples or kept fixed.
5. The manuscript contains multiple incomplete citations marked as “??”, e.g., on pages 6 and 11.
6. The primary results are based on ResNet-32 and ResNet-50. These architectures are relatively outdated; more modern alternatives like ViT or DINO should be considered to verify the generality.
7. The visualizations in Figure 3 are limited to CIFAR-10, which contains low-resolution images and relatively simple semantics. While the method is also evaluated on ImageNet-LT, a more challenging and high-resolution benchmark, no corresponding qualitative visualizations are provided.
8. The method adopts OT as the primary tool for distribution matching but lacks a clear rationale for why OT is preferred over other widely used metrics such as KL divergence, JS divergence, or MMD.

---

### Review · Reviewer_1nRv · 2025-04-25

**Summary Of Contributions:**

This paper proposes a method for addressing the long-tail classification problem, where standard training with cross-entropy results in poor performance on minority classes. The authors build on the M2m (Major to Minor) framework (Kim et al. 2020), which addresses class imbalance by synthesizing new minority samples from majority class examples. The method begins by selecting a sample from a majority class $k_0$, then translating it through an optimization procedure to create a new sample that resembles a target minority class $k$. This is done by encouraging a pretrained classifier $g$ to confidently classify the modified sample as class $k$, while also ensuring that a target classifier $f$ has low confidence in predicting it as the original class $k_0$.
​

The main contribution of this work is to regularize the sample synthesis objective by encouraging the distribution of generated minority samples to match the distribution of real minority samples in the training set. The paper also shows that this distribution-matching approach can be extended to synthesize minority samples from out-of-distribution (OOD) data and can be combined with other long-tail learning strategies, such as loss reweighting.

The authors evaluate their method on several long-tailed benchmark datasets (CIFAR10-LT, CIFAR100-LT, ImageNet-LT, and Places-LT). They report improved performance compared to existing methods. Additional ablation studies analyze the impact of each component in the objective, explore different distance metrics for matching distributions, and provide visualizations to illustrate how the method works.

**Audience:**

Yes

**Claims And Evidence:**

Yes

**Requested Changes:**

In addition to the issues raised above, there are some typos throughout the paper (some listed below). A careful proofreading pass would improve the paper.

- In Section 3.1 (second paragraph), it is stated that M2m optimizes the synthetic sample $ \hat{x} $ using gradient ascent. However, Equation (3) defines a minimization objective, so should this be gradient descent?
- *(minor)* In Section 4, the abbreviation IF (imbalance factor) is introduced without prior definition.
- At the end of page 6, a reference tag ("??") is missing when referring to a section.
- In Section 4.3, the expression $ L(k_c) $ should be defined as $ f_{k_c}(\hat{x}_m) $.
- At the end of page 11, there is another missing reference ("??") when pointing to a figure, and the results of section 4.8 seem missing.
- Section C.6, the same text is repeated twice.

**Strengths And Weaknesses:**

The paper proposes an effective and intuitive modification to the M2m framework for handling long-tailed classification. The idea is well-motivated, and the authors provide extensive experiments across multiple benchmarks. The method consistently outperforms existing approaches, and the ablation studies help demonstrate the contribution of each component in the objective. The visualizations and empirical breakdowns (e.g., using different distance metrics) are also helpful in understanding the method’s behavior.

However certain points need more clarification in the paper:

1.  The training framework described in the main body is somewhat unclear. Although the method builds on M2m, it would be helpful to include a more self-contained description to make the paper standalone. The proposed sample synthesis objective relies on the target classifier $f$, but it is not clearly explained how $f$ is trained jointly with the generation of new samples. The authors partly clarify this in the appendix that $f$ is first trained for 160 epochs before starting the sample synthesis (which I believe is worth mentioning briefly in the main body as well for clarity). However, it's still unclear how the joint updates of $f$ and sample synthesis are handled at each iteration.

2. The rejection condition labeled as "Reject = 1" in Algorithm 1 is not clearly explained within the algorithm itself and is only clarified later in the text, which can be confusing.  Also, when a sample is rejected, the method instead repeats existing minority samples. This fallback effectively acts as a form of loss reweighting, similar to class-based upsampling or weighted cross-entropy with weights inversely proportional to class frequencies. Reporting the proportion of synthesized versus repeated samples (e.g., in the setup of Figure 1(c)) would provide useful context for understanding how much of the improvement comes from the proposed synthesis mechanism as opposed to standard rebalancing strategies.

3. I believe, in section 4.4, $L(p)$ and $L(d)$ are used without prior definition. This makes the purpose of this section difficult to follow.

4.  While the paper includes a discussion on the computational cost of the proposed method in Section 4.8, the referenced figure appears to be missing. Moreover, the runtime comparison is limited to M2m, which only differs in the regularization term. For completeness, it would be helpful to include comparisons with other strong baselines—such as MetaSAug—that achieve close performance but follow different methodological approaches.

---

### Review · Reviewer_jj2N · 2025-05-11

**Summary Of Contributions:**

This paper addresses long-tailed classification where examples from many classes are only a few and only a few classes have many training examples. The authors take a familiar but well thought-out route in tackling these. The authors made a level playing field by generating samples which are less in number (minority samples). And these are made from majority samples. The formulation made sure the following.
 - synthetic samples are to be classified as the target minority class with high confidence. It should also have very low confidence to be classified as the source majority class.
 - The feature distribution of the synthesized samples and the original samples of the same class are made similar. The distance function used to measure similarity is the optimal transport distance.
 - Using the same distance criterion, the authors employ a sample rejection criterion where the distance of a synthetic minority sample from a real minority sample of the same class is high.
 - The authors employ another regularization term which accounts for the cases where minority samples are frequently misclassified into a specific class rather than other classes.

The authors motivated the choices and empirically validated them with adequate experiments. The analysis and ablation of the experiments made the work stronger.

**Audience:**

Yes

**Broader Impact Concerns:**

None as such.

**Claims And Evidence:**

Yes

**Requested Changes:**

If I have to make request for some change it would be mostly presentation related. Please refer to the first weaknesses I have mentioned above. At the same time, it would be good to have a small experiment on targeted embedding space by classical dimensionality reduction techniques like PCA or Probabilistic PCA and compare them to the results obtained by random embedding spaces.

**Strengths And Weaknesses:**

Strengths:
1. The presentation is generally good and motivations behind the choices of loss functions, regularizers and backbones are either logically or empirically validated.
2. Barring a few confusions, the Mathematical descriptions seem to be lucid.
3. Experiments are robust and rigorous, especially the ablations justifying the choices of the components. The experimental results suggest substantial improvement compared to the competing approaches.

Weaknesses:
1. The notations are confusing at certain places. E.g., a) While describing $\mathbf{C}$ just after eqn. (1), the use of $x_n$ and $y_m$ seems to be confusing. Would they be $u_n$ and $v_m$ instead of $x_n$ and $y_m$? I am asking as the distributions are defined in terms of $\mathbf{u}$ and $\mathbf{v}$ above the equation. b) (Last line of second paragraph of section 3.1): Is the condition $k_0 < k$ right? Or are they the number of examples in classes $k_0$ and $k$? c) The use of the acronym IF is never explained. It is first used in section 4. Does it mean Imbalance factor? What does it mean IF to be 4980/5? d) Results with the OOD Setting: In this subsection, it is never mentioned which table should we refer to look at the result. I feel it is table 2, but without proper reference, it is hard to judge the merit. e) The legends in Fig 1(a) and 1(b) are a little confusing when read along with the text in section 4.3. In the text, it is said that the method is compared with M2m with $L(k)+L(k_0)$, but in the legend there is no mention of M2m. So, it is hard to gauge whether M2m is employed or not. Additionally, the caption of Fig 1 says the IF value is 100 while the horizontal axes on each of the subfigures show 3 different IF values.
2. Section 3.3: Here different options of embedding spaces have been discussed and it has been argued that minimizing the distribution distance in the image space is expensive (section 3.2), thus an embedding function $\psi_\theta$ replaces the image space in computing the distribution distance. Though, in the experiments section, the experimental validation is provided for this, still details about the choice of the embedding function is missing. In section 3, it is only said that the feature embedding is obtained by first randomly sampling a randomly initialized network and then passing the images through it. How is the random initialization of the network done and how and why does it require to be resampled? On a related note, What about targeted ways of dimensionality reduction - say PCA or Probabilistic PCA on either the image space or feature space or parameter space? As this step is crucial in having the distribution distances, it is also important to understand, motivate and justify the choices made in this step.
3. Minor typos and issues: a) Section 2: Subsection Long-tailed classification: This description is pretty generic and applicable to general classification problem (including both long-tailed and non-long-tailed). So why is the subsection name 'Long-tailed classification' is not clear. b) Introduction second paragraph: ‘imbalanced issue’ -> ‘imbalance issue’; Section 3.2 – 4th paragraph: ‘highly dimensional’ -> ‘high dimensional’; Section 3.5 ‘for saving cost consumption’ -> ‘to reduce computation’; Page 6, last line - Section number missing; Page 11 last line – Figure number missing.

---

### Decision · Action_Editor_gYm5 · 2025-06-10

**Recommendation:** Accept as is

**Additional Comments:**

None

**Audience:**

Yes

**Audience Explanation:**

Learning from long-tailed class-imbalanced data is a challenging and relevant problem in the machine learning community.

**Claims And Evidence:**

Yes

**Claims Explanation:**

The authors propose a learning method from long-tailed class-imbalanced data by generating synthetic samples for minority classes, which contributes to improving the performance in experiments. However, since it was built on prior work, its novelty is rather limited. Also, theoretical analysis is left open.